

# 1 Assessing the cumulative impact of on-farm reservoirs on modeled
# 2 surface hydrology

Vinicius Perin[1*], Mirela G. Tulbure[2], Shiqi Fang[3], Sankarasubramanian Arumugam[3], Michele L.
Reba[4], Mary A. Yaeger[5]
[1] Planet Labs Inc., San Francisco, CA 94107, USA.
[2] Center for Geospatial Analytics, North Carolina State University, 2800 Faucette Drive,
Raleigh, NC 27606, USA
[3] Department of Civil, Construction and Environmental Engineering, North Carolina State
University, 961 Partners Way, Raleigh 27695, USA
[4] USDA-ARS Delta Water Management Research Unit P.O. Box 2, State University, AR 2467-
0002, USA.
[5] Center for Applied Earth Science and Engineering Research, The University of Memphis,
3675 Alumni Drive, Memphis, TN 38152, USA.
*Corresponding author: vperin@planet.com

## 16 Abstract

On-farm reservoirs (OFRs) are essential water bodies to meet global irrigation needs. Farmers
use OFRs to store water from precipitation and runoff during the rainy season to irrigate their
crops during the dry season. Despite their importance to crop irrigation, OFRs can have a
cumulative impact on surface hydrology by decreasing flow and peak flow. Nonetheless, there
is limited knowledge on the spatial and temporal variability of the OFRs' impacts. Therefore, to
gain novel understanding on the cumulative impact of OFRs on surface hydrology, here we
propose a novel framework that integrates a top-down data driven remote sensing-based
algorithm with physically-based models by leveraging the latest developments in the Soil
Water Assessment Tool+ (SWAT+). We assessed the impact of OFRs in a watershed located in



eastern Arkansas, the third most irrigated state in the USA. Our results show that the presence
of OFRs in the watershed decreased annual flow on average between 14 and 24%, and the
mean reduction in peak flow varied between 43 and 60%. In addition, the cumulative impact
of the OFRs was not equally distributed across the watershed, and it varied according to the
OFR spatial distribution, and their storage capacity. The results of this study and the proposed
framework can support water agencies with information on the cumulative impact of OFRs,
aiming to support surface water resources management. This is relevant as the number of
OFRs is expected to increase globally as an adaptation to climate change under severe
drought conditions.
## 1 Introduction
Inland water bodies (e.g., lakes and reservoirs) comprise a small fraction of Earth's surface;
however, they are responsible for storing the vast majority of the accessible fresh water
resources available on Earth. In addition, these water bodies are pivotal components of surface
hydrology, having key roles in ecosystem functioning and wildlife habitats (Khazaei et al., 2022;
Verpoorter et al., 2014). In particular, on-farm reservoirs (OFRs) are essential to meet global
irrigation needs (Döll et al., 2009; Downing, 2010; Van Den Hoek et al., 2019). Farmers use OFRs
to store water from precipitation and runoff during the rainy season to irrigate their crops
during the dry season (Habets et al., 2018; Perin et al., 2021; Vanthof & Kelly, 2019; Yaeger et al.,
2017; Yaeger et al., 2018). The number of OFRs is expected to rise worldwide in the coming
decades, and estimates show that there are more than 2.1 million OFRs in the US alone
(Downing, 2010; Renwick et al., 2005). OFRs are often built to manage surface water resources
more efficiently, and to help mitigate the impact of extreme droughts, which are projected to
increase due to climate change (Habets et al., 2018; Van Der Zaag & Gupta, 2008). Although
OFRs are small water bodies (< 50 ha), they can have cumulative impacts on the local and
remote hydrology in the watersheds where they occur (e.g., decreasing flow and peak flow)



(Habets et al., 2018), and their impact may contribute to worsening the surface water stress
already intensified by climate change and population growth (Vörösmarty et al., 2010). Most
studies have focused on the cumulative impact of major large reservoirs on downstream flow
alteration (Chalise et al., 2021; Mukhopadhyay et al., 2021), but limited analysis has been
performed on the impact of OFRs on downstream flow availability.

To quantify the impact of OFRs on surface hydrology, it is necessary to understand the

spatial and temporal variability of OFRs, as well as how the impacts are related to the OFR
networks, as the impacts of OFRs are not the sum of the individual OFR impacts, but rather
the sum and their interaction effects (Canter & Kamath, 1995; Habets et al., 2018). By gathering
information from several studies conducted in different countries (e.g., USA, France, Brazil),
Habets et al., (2018) did a thorough assessment of the OFRs' impact on surface hydrology, and
the different types of models and ways to represent the OFRs on the watershed. The authors
concluded that the modeled OFRs impacts have a wide range, and that most of the studies
reported a mean annual reduction in flow, which ranged between 0.2 and 36%. In addition, the
variability of the impact as identified in these previous studies was higher when assessing low
flows during multiple years, with reductions between 0.3 and 60%. In general, the estimated
mean annual reduction in flow was 13.4% ± 8.0%, and the mean decrease in peak flow was up
to 45% (Habets et al., 2018).

The approaches used to quantify the cumulative impact of OFRs can be divided into

two classes: data-driven methods, and process based hydrological modeling. The data-driven
approaches include three main methods. The first method relies on assessing measured
inflows and outflows of selected OFRs aiming to quantify their hydrological functioning with
the assumption that the cumulative impacts are the sum of individual impacts (Culler et al.,
1961; Dubreuil and Girard, 1973; Kennon, 1966). A variation of the cumulative impact assessment
approach has been recently suggested by Hwang et al., (2021) by comparing the naturalized
flows and the controlled flows for assessing the impact of large reservoir systems. The second
method is based on statistical analysis of the observed discharge time series of a watershed as



the number of OFRs increased (Galéa et al., 2005; Schreider et al., 2002). This approach is limited
when discriminating the specific impact of OFRs from those of land use and land cover
change, and when explicitly representing the OFRs in the models, given that OFRs tend to be
aggregated within the entire basin (i.e., OFRs surface area and/or storage are summed and
modeled as a unique water impoundment). The third method relies on conducting a paired-
catchment experiment by comparing the flows from two adjacent and similar catchments,
one with OFRs and the other without OFRs (Thompson, 2012). This technique requires the
catchment properties (e.g., soils, topology, lithology, land cover) to be spatially homogeneous,
which is practically nonexistent at a large scale, hence limiting this method's applications.
The second class of methods relate to hydrological modeling, and it is the most widely
used approach for assessing the OFRs' impacts. A variety of models have been proposed by
coupling the OFRs' water balance with a quantitative approach to estimate the OFRs' water
volume change (Fowler et al., 2015; Habets et al., 2014; Jalowska & Yuan, 2019; Yongbo et al.,
2014; Ni & Parajuli, 2018; Perrin, 2012; Zhang et al., 2012). In general, the models have three main
components: the OFR water balance, the quantitative approach to quantify the OFR inflows,
and the spatial representation of the OFRs network. These different model components result
in different limitations and assumptions—a complete assessment of these three components
and how they impact the hydrological simulations is provided in a recent review (Habets et al.,
2018). Therefore, when selecting a specific model to assess the impacts of the OFRs, it is
important to account for the model's suitability for the target issue to be addressed, as well as
the model limitations and assumptions. The selected model should also have capability to
incorporate/assimilate varying land-surface conditions (e.g., soil moisture) and time-varying
OFR storages which could be obtained either from local monitoring or through remote
sensing.
Most studies have used remotely-sensed products such as soil moisture (e.g., SMAP;
(Entekhabi et al., 2010), groundwater (e.g., GRACE; (Tapley et al., 2004) and land cover
conditions (e.g., MODIS; (Justice et al., 1998)) for assimilating current conditions into


hydrological models. Given that OFRs tend to occur in high numbers (e.g., hundreds), multiple
studies leveraged the latest developments and availability of satellite imagery to monitor the
occurrence and dynamics of OFRs (Jones et al., 2017; Ogilvie et al., 2018, 2020; Perin et al., 2022;
Perin et al., 2021a, 2021b; Van Den Hoek et al., 2019; Vanthof & Kelly, 2019), which could provide
useful information on local storage conditions for predicting downstream streamflow. Further,
these studies allowed quantifying the number of OFRs, and their spatial and temporal
variability in surface water area and storage in the watershed where they occur, providing
relevant information when modeling the cumulative impact of OFRs. Despite the
complementary information provided by satellite imagery, there are only a few studies that
incorporated remote sensing-derived information (e.g., soil moisture derived from SMAP,
groundwater based on GRACE) with hydrological modeling (Ni and Parajuli, 2018; Yongbo et
al., 2014; Zhang et al., 2012), and these studies are limited to mapping the OFRs occurrence, or
to snapshots of the OFRs conditions (e.g., surface area). To the best of our knowledge, there is
no study that combines the spatial and temporal variability of the OFRs—derived using multi-
year satellite imagery time series analyses—with a process-based hydrological model.
Therefore, to gain novel understanding of the cumulative impact of OFRs on surface
hydrology, in this study, we propose a new approach that systematically integrates the
dynamically varying conditions of OFRs based on satellite imagery time series (Perin et al.,
2022) using a top-down data driven approach within the latest SWAT+ model. The Soil and
Water Assessment Tool (SWAT) (Arnold et al., 2012) has been widely used to model the impacts
of the OFRs (Jalowska and Yuan, 2019; Kim and Parajuli, 2014; Ni et al., 2020; Ni and Parajuli,
2018; Perrin, 2012; Rabelo et al., 2021; Yongbo et al., 2014; Zhang et al., 2012), in part given by a
comprehensive collection of model documentation and guidelines available online
(https://swat.tamu.edu/). Our objectives are to (1) assess the spatial and temporal variability of
the cumulative impact of OFRs at the watershed and subwatersheds levels, and (2) to quantify
the intra- and-inter annual impacts of the OFRs on flow and peak flow at the channel scale. By
integrating the SWAT+ model with a novel remote sensing assimilation algorithm to account



for the OFRs spatial variability—which is lacking in most of studies assessing the OFRs
impacts—and leveraging a digitally-mapped OFRs dataset (Yaeger et al., 2017), we are
providing a new approach that can be replicated in watersheds across the world, and used to
support water agencies with information to improve surface water resources management.
## 2 Methods
### 2.1 Study region
The study region is located in eastern Arkansas, USA, the third most-irrigated state in the USA
(ERS-USDA, 2017). The area has a humid subtropical climate with a 30-year annual average
precipitation of ~1300 mm/year (PRISM Climate Group, 2022). The precipitation is distributed
mostly between March and May, receiving an average of ~400 mm during these months (Perin
et al., 2021b). The region has experienced a steady increase in irrigated agriculture, with
commonly irrigated crops including corn, rice, and soybeans (NASS-USDA, 2017). A recent
study (Yaeger et al., 2017) digitally mapped 330 OFRs located in the study region (Fig. 1) using
the high-resolution (1-m) National Agricultural Imagery Program archive in combination with
2015 sub-meter spatial resolution Google Earth satellite imagery. Most of the OFRs (95%) have
surface area < 50 ha, and they are concentrated in the eastern portion of the study region (Fig.

1).

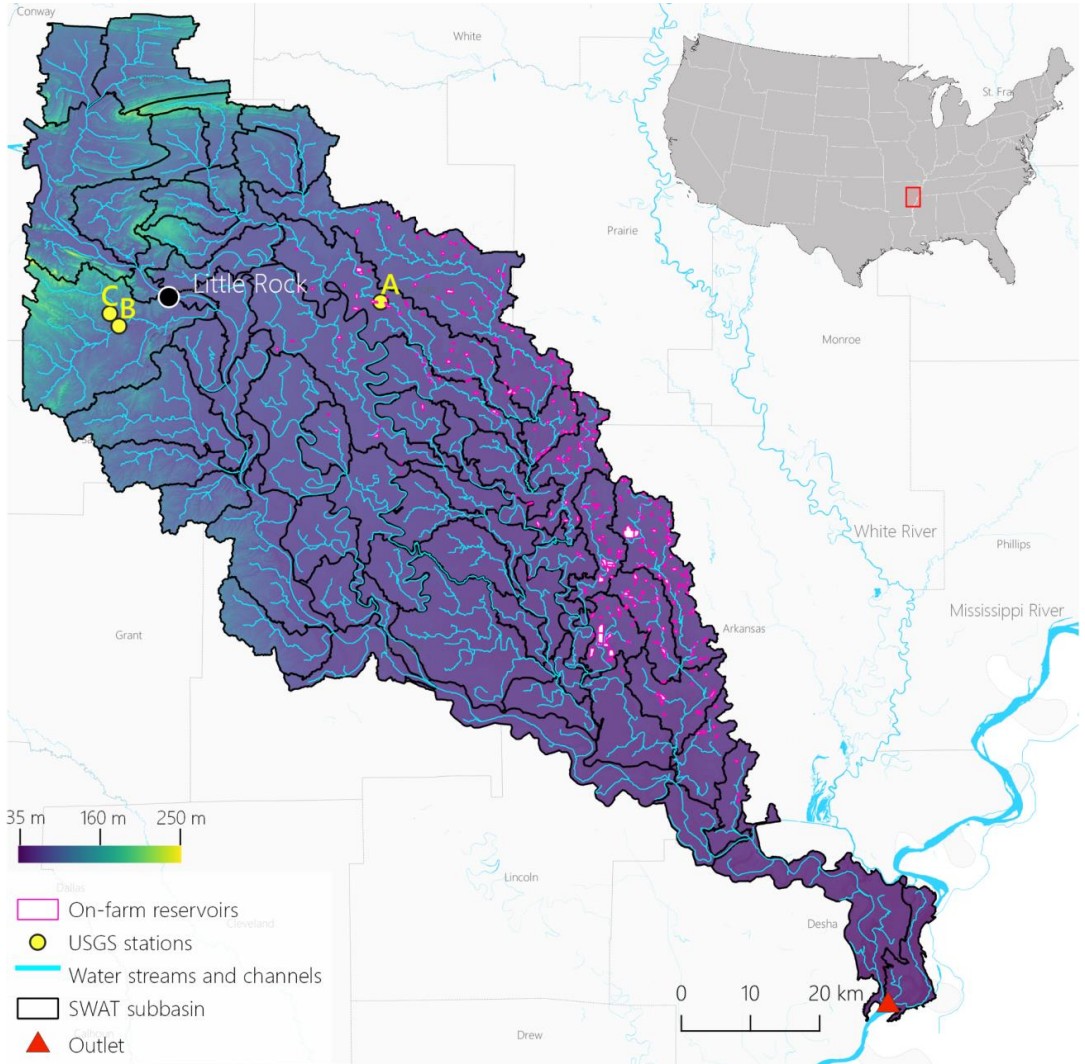

**Figure 1**–Study region located in eastern Arkansas, USA, the subwatersheds and surface water streams and channels delineated with SWAT+, the model outlet, the United States Geological Survey (USGS) stations (United States Geological Survey Water Data for the Nation, 2022) used for flow calibration and validation, the digitized OFRs (Yaeger et al., 2017), and the Digital Elevation Model (DEM) used in the modeling (Farr et al., 2007).



2.2 SWAT+ model setup
2.2.1 The Soil Water Assessment Tool to model the impacts of OFRs on surface hydrology
The SWAT model is a time-continuous semi-distributed hydrological model widely used across
the globe—more than 5,000 peer reviewed publications since its launch in the early 1980s
(Publications | Soil & Water Assessment Tool (SWAT), 2022). The large number of SWAT
applications globally revealed the model development needs and its limitations. To address
the present and future challenges when modeling with SWAT, the model source code has
undergone major modifications, and a completely revised version of the model was proposed
in SWAT+ (Bieger et al., 2017). SWAT+ uses the same equations as SWAT to simulate the
hydrological processes; however, it offers more flexibility to users when configuring the model
(e.g., when defining management schedules, routing constituents, and connecting managed
flow systems to the natural stream network) (Bieger et al., 2017).
The SWAT+ is under constant improvements (Chawanda et al., 2020; Molina-Navarro et
al., 2018), and a new module (Molina-Navarro et al., 2018) was recently developed to allow the
optimal integration of a water body and its drainage area within the simulated hydrological
processes. In previous versions of the model, when delineating the watershed area draining
into a water body, the users were required to place an outlet in a certain point of the water
stream's network, and the areas in-between the rivers' subwatersheds flowing into the water
body were therefore excluded—if these areas are disregarded, important hydrological
processes (e.g., evaporation, overland and/or groundwater flow) flowing into the water body
are not accounted for (Molina-Navarro et al., 2018). This former approach can lead to
inaccuracies when delineating the watershed areas, especially when the results are used as
input to an OFR model component. The newest versions of SWAT+ consider the OFRs' outline
(i.e., shape and surface area) when delineating the watersheds; hence, accounting for the
entire drainage area flowing into the waterbody (Mollina-Navarro et al., 2018). In addition, the
latest versions allow adding more than one OFR per subwatershed by associating the OFR


with channels—components of the watersheds, and finer divisions and extensions of water
stream reaches—enabling the modeling analyses at the channel scale. When simulating the
impact of the OFRs at the channel scale, there is a higher level of detail of where and when the
OFRs are contributing to changes in surface hydrology, unlike the previous versions of the
model, which allowed adding only a single OFR per subwatershed placed at the subwatershed
outlet as a point (Arnold et al., 2012), and therefore, the analyses were conducted at the
subwatershed scale.
We modeled the impact of OFRs on surface hydrology using the QSWAT+ (v.2.1.9)
SWAT+ model interface together with SWAT+ Editor (v.2.1.0) to set up the model, to input the
required datasets (e.g., DEM, land use and land cover layer, interpolated meteorological
climate information), and to run the different modeling scenarios.
The modeled watershed (710,700 ha, Fig. 1) included 68 subwatersheds and a total of
642 Hydrological Response Units (HRUs)—HRUs are unique portions of the subwatersheds
that have unique land use and management, and soil attributes. We set up daily simulations
for 30 years (1990–2020), including five years of model warm up to establish the initial soil water
conditions and hydrological processes. The watershed was delineated using the Shuttle Radar
Topography Mission DEM (30 m) (Farr et al., 2007). In addition, we set the channel length
threshold to 6 km$^2$, and the stream length threshold to 60 km$^2$. We placed an outlet in the
southern part of the study region—where the lowest part of the watershed is located (Fig. 1).
We created the HRUs using the dominant option—this option selects the largest HRU within
the subwatershed as the general HRU—within QSWAT+ interface, and used the National Land
Cover Database (30 m) (Homer et al., 2020), and Gridded Soil Survey Geographic Database
(gSSURGO) (Soil Survey Staff, USDA-NRCS, 2021) (100 m) as inputs to the model. The gSSURGO
layers were processed according to their guidelines when using them on QSWAT+ (George,
2020). For climate data, we extracted the centroid coordinates of each subwatershed (Muche
et al., 2020), and used these centroids to download 30 years of daily precipitation, minimum
and maximum temperature, surface downward shortwave radiation, wind velocity, and



relative humidity from the Gridded Surface Meteorological Datasets (Abatzoglou, 2013),
available in Google Earth Engine (Gorelick et al., 2017). The time series of each subwatershed
centroid was added into the SWAT+ Editor as independent weather stations.

### 2.2.2 Model calibration and validation procedures

We used monthly measured flow from three USGS stations (Fig. 1 and Table 1) to calibrate and
validate the model flow simulations. The USGS flow time series length varied between 14 and
25 years, and we used 60% of the timeseries for calibration and 40% for validation for each
USGS station (Table 1). We assessed the performance of the model by calculating the
Coefficient of determination ($r^2$), Percent bias (PBIAS, %, Eq. 1) (Yapo et al., 1996), and the Nash–
Sutcliffe model efficiency coefficient (NSE, Eq. 2) (Nash and Sutcliffe, 1970). PBIAS is the relative
mean difference between the simulated and the measured flow values, and it reflects the
ability of the model to simulate monthly flows. The optimal PBIAS is zero, and low-magnitude
values indicate better model performance. Positive PBIAS indicates overestimation bias,
whereas negative values denote underestimation bias. The NSE expresses how well the model
simulates flows, and it ranges from a negative value to one, with one indicating a perfect fit
between the simulated and measured flow values. In general, the model simulations of
monthly flow are considered satisfactory when $r^2$ ranges from 0.60 to 0.75, PBIAS ranges from
±10% to ±15%, and NSE ranges from 0.50 to 0.70 (Moriasi et al., 2015).
**Table 1**–USGS stations, drainage areas, and the periods used for flow calibration and validation.

| USGS station | Station id | Drainage Area (ha) | Period (years) | |
|---|---|---|---|---|
| | | | Calibration | Validation |
| 07264000 | (A) | 53,600 | 1995–2010 | 2010–2020 |
| 07263555 | (B) | 25,400 | 2007–2014 | 2014–2020 |
| 07263580 | (C) | 5,300 | 1997–2011 | 2011–2020 |






PBIAS $= \frac{\sum_{i=1}^{n} (Yi - Xi)}{\sum_{i=1}^{n} Xi}$

(1)

NSE $= 1 - \frac{\sum_{i=1}^{n} (Xi - Yi)^2}{\sum_{i=1}^{n} (Xi - \underline{Xi})^2}$            (2)
Where $X_i$ is the measured flow and $Y_i$ is the simulated flow.
We conducted a sensitivity analysis using the SWAT+ ToolBox (v.0.7.6) (SWAT+ Toolbox,
2022) to reveal the most sensitive parameters when simulating flow—a total of 10 parameters
(Table S 1) were tested based on previous studies that used SWAT/SWAT+ to model the impact
of water impoundments on surface hydrology (Jalowska & Yuan, 2019; Yongbo et al., 2014; Ni et
al., 2020; Ni & Parajuli, 2018; Perrin, 2012; Rabelo et al., 2021; Zhang et al., 2012). Following the
sensitivity analysis, we selected the five most sensitive parameters (Table 2), and proceeded
with a manual calibration using the SWAT+ Toolbox. We aimed to improve the model's
monthly flow predictions by testing the parameters one at a time and changing their values
between -20% to 20% with 5% increments based on their range values. The final calibrated
parameters and their fitted values are shown in Table 2.
**Table 2**–Monthly flow calibration parameters.

| Parameter | Description | Range | Value |
|-----------|-------------|-------|-------|
| CN2 | SCS runoff curve number | 35–95 | 0.20* |
| SOL_AWC | Available water capacity (mm/mm) | 0.01–1 | -0.20* |
| ESCO | Soil evaporation compensation coefficient | 0.01–1 | 0.50 |
| PERCO | Percolation coefficient (fraction) | 0–1 | 0.60 |
| CANMX | Maximum canopy storage (mm) | 0–100 | 75 |

*Denotes relative percentage change.
2.3 OFRs representation in SWAT+
Multiple OFRs can be added to the same subwatershed by associating them with channels
(Dile et al., 2022). The OFRs need to have at least one outlet channel, and they may have none
or multiple inlets. Therefore, most OFR-related processes within the model involve





determining what channels form inflowing and outflowing channels for each OFR. Ideally,
each OFR would interact with a channel, and therefore, have a channel entering, leaving, or
within the OFR. Nonetheless, it is common to have OFRs that do not intersect with any channel
(Dile et al., 2022)—this is the case for 93% of the OFRs in our study region. The OFRs from our
study region are not dammed along the streams, but rather they are engineered water
impoundments that are indirectly connected to the main streams via pipes and pumps
(Yaeger et al., 2017). A possible solution would be modifying the OFRs' shapes by dragging
them to the closest channel (Dile et al., 2022). However, this would require extensive
modifications of the OFRs' shapes. In addition, when an OFR is added to a channel, this channel
is split into two channels, and the model needs to account for the two newly created channels
during the water routing calculations. For this reason, adding multiple OFRs to the same
channel, or adding multiple OFRs closely located to the same channel, can be a cumbersome
process that leads to numerous routing errors.

To overcome these challenges, we aggregated the OFRs' surface area, and added

aggregated OFRs to the model. This adaptation involved two steps. First, for each of the 330
OFRs, we searched for the closest channel by calculating the distance between the OFRs'
centroid and the multiple channels within each subwatershed. Then, we aggregated all the
OFRs that were associated with each channel by summing up their surface area, and adding
a polygon of the aggregated area to represent the aggregated OFR. This approach resulted in
69 aggregated OFRs that were added to 67 different channels located in 16 subwatersheds.
The surface area of the aggregated OFRs varied between 3.05 ha and 165.67 ha, and the
number of OFRs in each aggregated OFR varied between 2 and 12. To avoid confusion, for the
rest of the manuscript, we refer to OFRs as the aggregated OFRs, and not the individual OFRs
shown in Fig. 1. For each of the aggregated OFR, the water volume was calculated using SWAT+
default rule, which is a simple multiplication of the OFR surface area by a factor of 10, similar
to other studies based on SWAT+ (Ni and Parajuli, 2018; Zhang et al., 2012). In addition, given that
we did not have access to the OFRs release rates, we used the model default release rule, which





sets the OFRs to release water when the spillway volume is reached—80% of the OFRs capacity
(Bieger et al., 2017).
2.4 Scenario Analysis
Given our representation of the OFRs in SWAT+, we assessed the impact of the OFRs on surface
hydrology at the channel scale. To do so, we established the model baseline scenario without
the presence of the OFRs on the watershed. In addition, we divided the channels into four
classes (i.e., low and high flow classes) according to their mean baseline flow. The different class
intervals were calculated using the mean flow quartiles accounting for all channels, which
resulted in the following baseline flow classes: (1) 0.001–0.25 $m^3$/s, (2) 0.25–0.50 $m^3$/s, (3) 0.50–2.11
$m^3$/s, and (4) 2.11–17.50 $m^3$/s.
To account for the OFRs variation in surface area (i.e., change in storage capacity), we
propose a novel approach that leverages a top-down data-driven model based on satellite
imagery (Fig. 2). We used this model to create three modeling scenarios using daily OFRs
surface area time series—these scenarios were based on the methodology proposed by Perin
et al., (2022). The authors used a multi-sensor satellite imagery approach with the Kalman filter
(Kalman, 1960) to derive daily OFRs' surface area change between 2017 and 2020. The proposed
algorithm accounts for the uncertainties in both the sensor's observations and the resulting
surface areas. By improving the OFRs surface area observations cadence, the algorithm allows
further understanding of the OFRs surface area intra- and inter-annual changes, which are key
pieces of information that can be used to better assess and manage the water stored by the
OFRs (Perin et al., 2022). The daily surface area time series—derived by combining PlanetScope,
RapidEye, and Sentinel-2 satellite imagery (Perin et al., 2022)—of each OFR was used to
simulate three scenarios (i.e., lower, mean, and upper) representing the OFRs' capacity in terms
of surface area. The mean scenario represents the regular condition of the OFRs, and it is the
mean of the daily surface area time series derived from the Kalman filter. The lower and upper
scenarios represent the lowest and highest capacities of the OFRs, and they are based on the



surface area 95% confidence interval limits, calculated using the daily time series. For each
scenario, the OFRs were simulated at full capacity (i.e., maximum storage at the lower, mean
and upper scenarios), and this capacity was kept constant during the simulation period (Ni et
al., 2020; Ni and Parajuli, 2018; Perrin, 2012). To assess the impact of the OFRs on surface
hydrology, we compared the baseline flow with the flow simulated by each surface area
scenario—i.e., comparing the flow changes with and without OFRs, a common approach used
by previous studies (Habets et al., 2018).

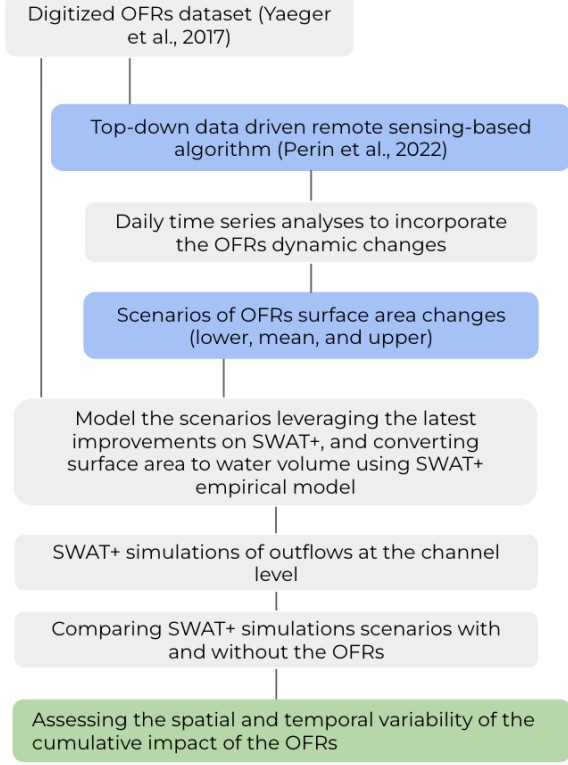


**Figure 2**–A new approach to integrate a top-down data driven remote sensing-based
algorithm, that assesses the OFRs dynamic conditions (Perin et al., 2022), with the latest SWAT+
model developments.
We estimated the impact of the OFRs on surface hydrology by calculating the percent
change (Eq. 3) of monthly flow between the baseline and the three surface area scenarios





including all OFRs. The annual impact on flow was calculated by averaging the mean percent
change along the months. We also calculated the distribution of the percent change for each
baseline flow class. The distribution was assessed using 2-D Kernel Density estimation (KDE)
plots. Different from discrete bins (e.g., histograms), the KDE plots show a continuous density
estimate of the observations using a Gaussian kernel. In addition, we assessed the percent
changes in peak flow. For the purposes of this analysis, peak flow is defined as equal or higher
than the 99[th] flow percentile calculated using the entire flow time series (Eq. 3).
Percent change (%) = $\left(\frac{Y_i - X_i}{X_i}\right) * 100$                                   (3)
Where $X_i$ is the baseline flow and $Y_i$ is the simulated flow of each surface area scenario.

## 3 Results

### 3.1 Model calibration and validation

The model calibration and validation were done using the three USGS stations presented in
Fig. 1 and Table 1, and accounting for all OFRs in study region. When comparing the monthly
simulated flow with the measured flow for the calibration period, there was a good agreement
($0.71 \leq r^2 \leq 0.93$), and a satisfactory model efficiency ($0.68 \leq NSE \leq 0.90$) for all three stations
(Fig. 3). In addition, the PBIAS magnitude was < 3% for station A, and < 12% for stations B and C.
Meanwhile, the validation period had $r^2$ ranging between 0.69 and 0.86, and the NSE between
0.68 and 0.83, with PBIAS magnitude < 10% for stations A and B, and 18.12% for station C. In
general, for stations A and C, the model overestimated flow values (i.e., positive PBIAS) mostly
during flow events < 3 m³/s, and the model underestimated flow (i.e., negative PBIAS) for
station B during flows > 20 m³/s (Fig. 3). These findings are consistent with a previous study
conducted in western Mississippi near our study region (Ni and Parajuli, 2018). Even though
during the validation period the station B had PBIAS magnitude higher than 15%, the $r^2$ and
NSE values from the calibration and validation periods indicate satisfactory modeling
performance when simulating monthly flow (Moriasi et al., 2015). Given that none of the OFRs





were directly connected with the streams where the stations were located (Fig. 1), and there
were no OFRs nearby stations B and C, the calibration and validation metrics with and without
the OFRs were very similar, with differences smaller than 1%.


**Figure 3**–Flow calibration and validation time series for the three USGS stations A (07264000),
B (07263555) and C (07263580). See Fig. 1 and Table 1 for more information about the USGS
stations. The precipitation time series represents the monthly accumulated precipitation at
the watershed scale (i.e., for the entire study region).
3.2 Percent change in flow
We assessed the impact of the OFRs on flow by comparing the baseline flow (i.e., without the
OFRs) with the three surface area scenarios generated from the Kalman filter approach—
lower, mean, and upper (see section 2.4, and Fig. 2). The total surface area (i.e., summing all
OFRs surface area) was 2.176 ha for the lower, 2.766 ha for the mean, and 3.370 ha for the upper,
and the three scenarios had a similar OFRs surface area distribution (Fig. 4). In addition, most
of the OFRs had surface areas < 50 ha—78%, 71%, and 62% of the OFRs for the lower, mean, and
upper scenarios.

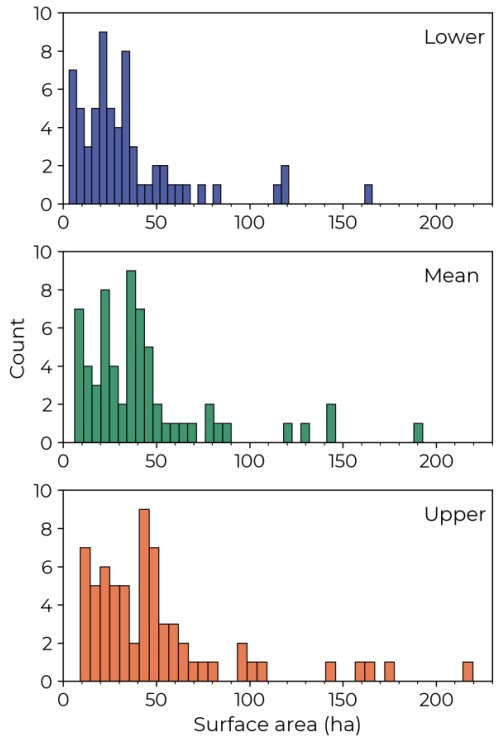




**Figure 4**–OFRs surface area distribution for the three surface area scenarios, lower, mean,
and upper.

The impact of the OFRs on monthly flow varied throughout the year, and the largest

impacts occurred between January and May for all flow classes (Fig. 5). During these months,
including all surface area scenarios, the mean decrease in flow (i.e., negative mean percent
change) was -34.4 ± 6% for class 1, -37.6 ± 5% for class 2, -30.0 ± 6% for class 3, and -34.1 ± 6% for
class 4. For all classes, the greatest reduction in flow occurred during the month of March (~ -
40%). Meanwhile, the impact of the OFRs was smaller during the second half of the year, in
which the mean percent change in flow was -12.0 ± 3.% for class 1, -12.5 ± 5% for class 2, -1.4 ± 4%
for class 3, and -2.6 ± 10% for class 4 (Fig. 5). So we always saw a decrease? It looks like we have
some increases too.

When assessing the mean percent change per month, for all surface area scenarios,

the lower flow classes (i.e., (1) 0.001–0.25 m³/s and (2) 0.25–0.50 m³/s) had a negative mean
percent change for all months. Nonetheless, we observed a mean positive percent change (i.e.,
increase in flow) for the months of August (5.0 ± 1%) and October (5.2 ± 0.2%) for class 3, and
during June (8.2 ± 0.3%), August (7.3 ± 0.4%), and October (8.7 ± 0.4%) for class 4 (Fig. 5).
Furthermore, the different surface area scenarios had similar impacts on flow for all months of
the year with differences smaller than 5% for all scenarios. Between January and May, for all
flow classes, the mean percent change was -32.0 ± 6% for the lower, –34.6 ± 7% for the mean,
and -35.8 ± 5% for the upper. Between June and December, the impact on flow was -5.4 ± 6%
for the lower, -7.3 ± 8% for the mean, and -8.9 ± 5% for the upper.


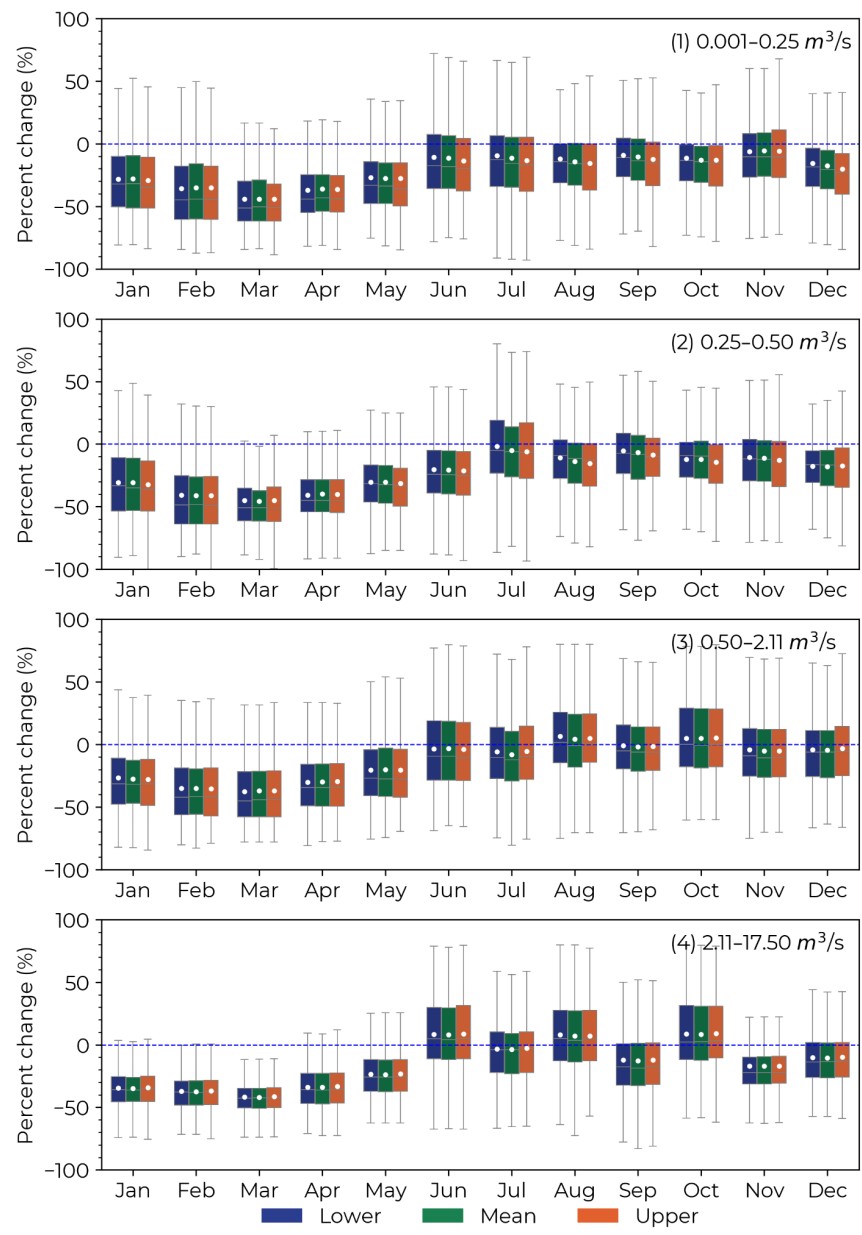

**Figure 5**–Monthly percent change in flow between the baseline scenario (vertical dotted blue
line) and the three surface area scenarios (lower, mean, and upper), and for the four flow classes
(1) 0.001–0.25 m³/s, (2) 0.25–0.50 m³/s, (3) 0.50–2.11 m³/s, and (4) 2.11–17.50 m³/s.





In general, the OFRs contributed to decreased monthly flow. However, the OFRs'
impact on flow had a significant intra- and inter-annual variability, and it varied according to
different OFRs and channels—this is highlighted by the boxplots size variability in Fig. 5, in
which the variability was lower during the first part of the year, and greater between July and
August. In addition, the monthly percent change in flow in the KDE plots (Fig. 6) shows that
for the three scenarios, and all flow classes, most of the changes in flow ranged between -40%
and 0%. In addition, all KDE plots have a triangular shape with its base on the smaller flows,
denoting where most of the changes occur. Even though the majority of the percent change
in flow is negative, there are circumstances in which the OFRs could positively impact flow—
the increase in flow is represented by faded colors in each surface area scenario (Fig. 6). The
positive mean percent change could be as high as 80%—see Fig. 6 for the larger flow classes,
(3) 0.50–2.11 m$^3$/s and (4) 2.11–17.50 m$^3$/s. The positive impact on flow for these classes occurred
during the months of June, August and October when a mean positive change is observed
(Fig. 5 classes 3 and 4).
The annual mean percent change, for all surface area scenarios, was -22.5 ± 3% for class
1, -24.2 ± 4% for class 2, -14.6 ± 3% for class 3, and -16.6 ± 3% for class 4. In addition, the surface
area scenarios annual changes were -18.0 ± 5% for the lower, -19.6 ± 5% for the mean, and -20.8
± 6% for the upper, including all flow classes. The differences between the surface area
scenarios shown in Fig. 5 and Fig. 6 are related to the variability of the OFRs surface area.




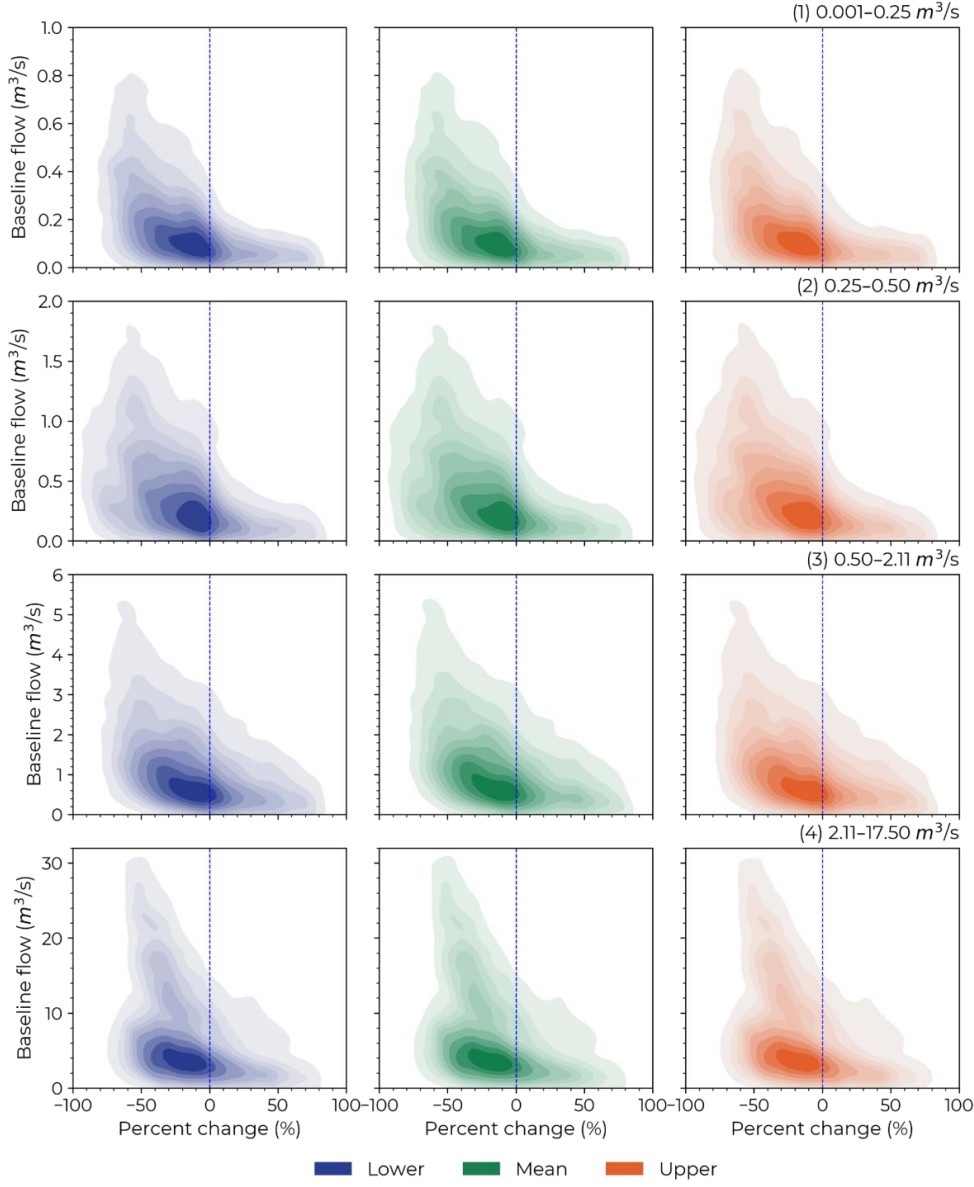


**Figure 6**–Kernel density estimation plots smoothed using a Gaussian kernel for the monthly

percent change in flow between the baseline scenario (vertical dotted blue line) and the three

surface area scenarios (lower, mean, and upper) for the four flow classes (1) 0.001–0.25 m³/s, (2)

0.25–0.50 m³/s, (3) 0.50–2.11 m³/s, and (4) 2.11–17.50 m³/s. Note the different range of values on

the y-axis for all four flow classes.



### 3.3 Impact on peak flow


For each channel, we calculated the impact of the OFRs on peak flow (Fig. 7). The impact on
peak flow was -60.7 ± 13% for class 1, -56.2 ± 11% for class 2, -46.7 ± 19% for class 3, and -43.9 ± 12%
class 4. When assessing the impact on peak flow based on different surface area scenarios, the
mean percent change was -49.4 ± 18% for the lower, -50.4 ± 17% for the mean, and -52.7 ± 18%
for the upper. All peak flows occurred between January and May, which is the period of the
year when the study region receives most of its precipitation (Perin et al., 2021). With the
exception of a few outliers, there was no increase in peak flow, even though the OFRs
contributed to a positive mean percent change in flow in certain months of the year (Fig. 5
classes 3 and 4).

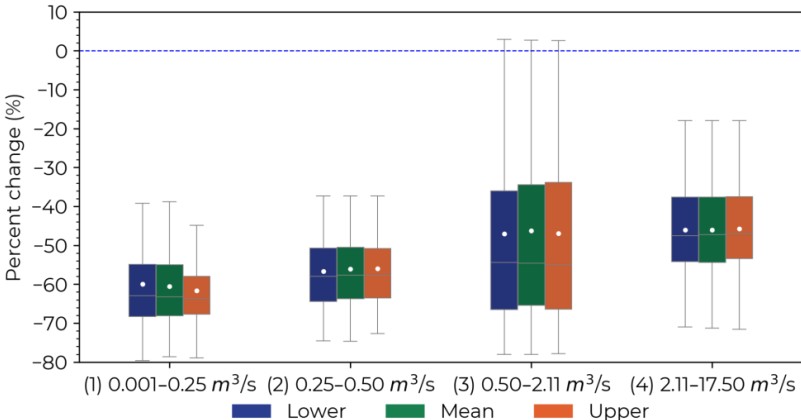


**Figure 7**–Percent change in peak flow between the baseline scenario (vertical dotted blue line)
and the three surface area scenarios (lower, mean, and upper) for the four flow classes (1) 0.001–
0.25 m³/s, (2) 0.25–0.50 m³/s, (3) 0.50–2.11 m³/s, and (4) 2.11–17.50 m³/s.

### 3.4 Simulated flow time series


We randomly selected a channel within the flow class 3 to demonstrate the baseline and the
three surface area scenarios' flow time series between 1995 and 2005 (Fig. 8). For this channel,
the annual mean percent changes in flow when comparing the baseline scenario with the
lower, mean, and upper surface area scenarios were 0.99 ± 11.8%, -1.9 ± 13%, and -2.0 ± 19%—the





high standard deviation for the three scenarios is explained by the interannual variability. The
upper surface area scenario resulted in lower flows (i.e., higher impact) when compared to the
lower and mean scenarios for the majority of the flow events—67.8% and 57.6% for the lower
and mean scenarios. Nonetheless, there are circumstances when the upper scenario yielded
higher flows—32.2% and 42.4% of the events for the lower and mean scenarios (e.g., see the
two insets 03/1997–08/1998 and 05/2002–02/2004). These findings indicate that the impacts
that the OFRs have on flow are not entirely governed by the presence and surface area of the
OFRs (i.e., the different surface area scenarios), instead by a combination of the OFRs with
different modeling components (e.g., terrain, land use, climate information), and different
hydrological processes (e.g., run-off, precipitation, evaporation). In addition, the impact on
peak flow for this channel was -45.7 ± 19.7% for all surface area scenarios—this is highlighted
on two occasions (08/2002 and 08/2003) during the second inset.

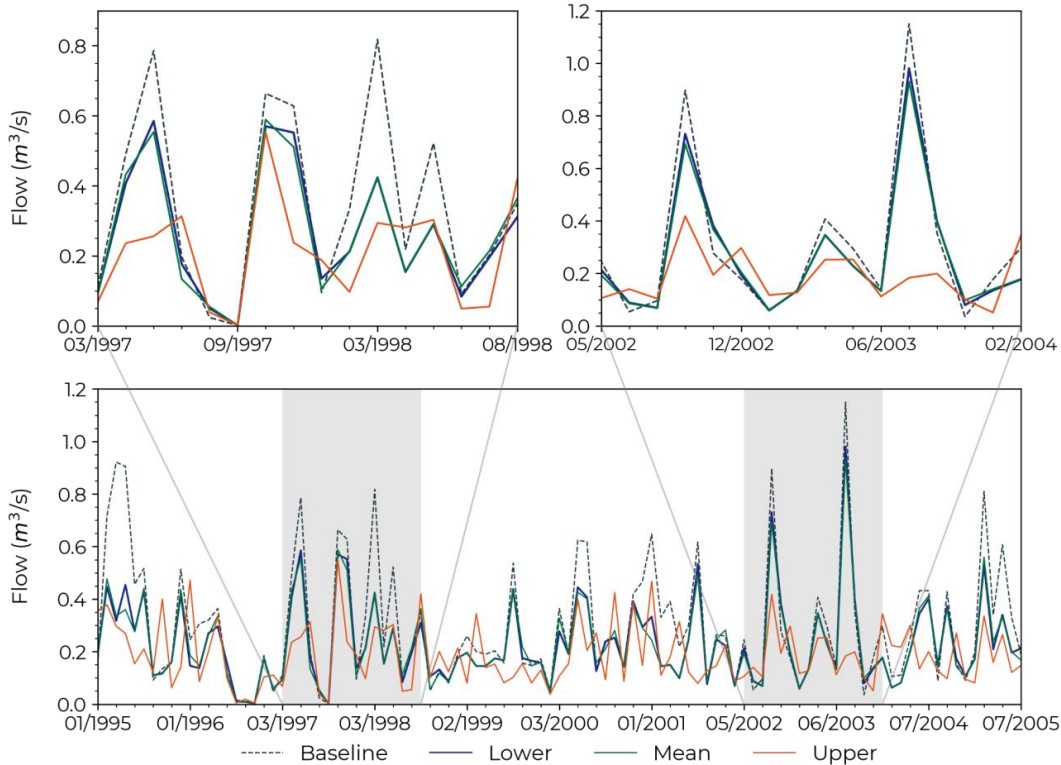






**Figure 8**–A subset of the time series of simulated flow for baseline and the three surface area
scenarios (lower, mean, and upper) between 1995 and 2005 for a selected channel within the
flow class 3.











3.5 Overall impact of OFRs

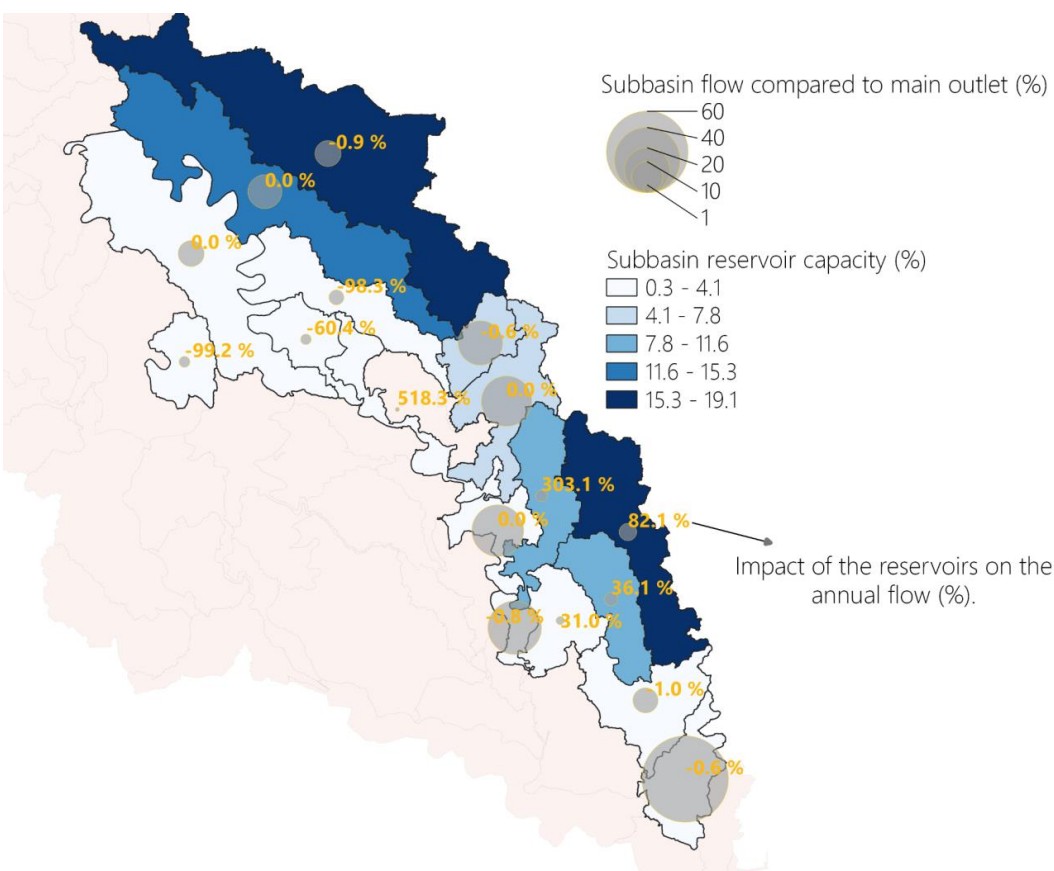


**Figure 9**–The cumulative impact of OFRs on annual flow for the mean scenario at the
subwatersheds where the OFRs occurred. The size of the circles represents the contribution
(%) of the subwatershed flow compared to the main outlet (i.e., model outlet). The
subwatersheds are color coded according to their reservoir capacity (%), which was calculated
by summing the OFRs surface area in each subwatershed and dividing the sum to the total
OFRs surface area (i.e., including all OFRs from all subwatersheds), darker color indicating
higher reservoir capacity. The percentages highlighted in yellow represent the impact of the
OFRs on annual flow.



To assess the overall impact of the OFRs at the subwatershed level, we calculated the
contribution of each subwatershed flow to the main model outlet, and the subwatersheds'
reservoir capacity (i.e., summing the OFRs surface area at each subwatershed and dividing it
to the total OFRs surface area, including all OFRs from all subwatersheds) (Fig. 9). In general,
the highest impacts on annual flow (e.g., > 100%), with positive or negative magnitude,
occurred at the subwatersheds that contributed the least (< 10%) to the main model outlet—
these subwatersheds are represented in lighter shades of blue, and the annual impact is
highlighted in yellow on Fig. 9. In other words, the highest impacts on flow occurred on the
channels with smaller flow magnitudes (e.g., channels that presented mean flow ranging
between 0.001–0.25 and 0.25–0.50 $m^3$/s, these channels were classified as class 1 and 2 in this
study). In addition, the subwatersheds with the highest reservoir capacities (between 15.3 and
19.1 %, represented in darker shades of blue) (Fig. 9), had a small (< 10%) contribution to the
model outlet, and these subwatersheds did not present the highest impact on annual flow
(e.g., the impact on annual flow for the top two subwatersheds in terms of reservoir capacity
were -0.9 and 82.1%).



## 4 Discussion


Although OFRs will contribute to improve food production resilience—by providing surface
water to irrigation during dry periods—to severe drought events, which are  expected to have
higher occurrence with climate change, OFRS can have cumulative impacts on surface
hydrology of the watershed where they occur. Studies have either used data driven or
physically based hydrological model approaches to estimate OFR impacts on the watersheds,
despite the fact that combining the two approaches leads to better understanding on what is
the spatial and temporal variability of the OFR impacts, given that the dynamic changes of the
OFRs are incorporated into the hydrological model. To quantify whether the impact of the
OFRS on mean and peak flow varied intra- and inter-annually, and which subwatersheds are
more impacted, here we combined a data-driven remote sensing-based model with SWAT+
latest improvements to assess the OFR impacts.
### 4.1 Cumulative impact of OFRs

When simulating water impoundments in SWAT/SWAT+, it is common practice to

validate and calibrate the model using flow measurements (Evenson et al., 2018; Habets et al.,
2018; Jalowska & Yuan, 2019; Ni & Parajuli, 2018). In addition, other studies have validated and
calibrated the model using alternative variables. For example, Perrin et al., (2012) employed
monthly measurements of piezometric variations to assess aquifer recharge processes, and
Jalowska & Yuan (2019) used sediment loadings (concentration and budget), from field
monitoring reports to evaluate sediment simulations. Ideally, we would calibrate and validate
the model by accounting for the parameters governing the OFRs' water budget (e.g., inflows
and outflows) (e.g., Kim and Parajuli, 2014). Nonetheless, these measurements were not
available for the OFRs in our study region. Furthermore, a thorough calibration and validation
of the model would require extra flow data, covering other parts of the study region, as the
three USGS stations—the only data available—used in this study are located in the upper part


of the modeled watershed. Similar to Evenson et al., (2018)—who proposed a module to better
represent spatially distributed wetlands, and validated their model using a direct (i.e., flow
measurement) and an indirect (i.e., the wetlands surface area) approach—our validation and
calibration was done using the flow measurements, and the OFRs surface area scenarios were
based on an algorithm that was validated with an independent higher spatial resolution
dataset (Perin et al., 2022).
There is a consensus within the scientific community that the OFRs will have a
cumulative impact on surface hydrology by decreasing flow and peak flow, and the impact will
vary from watershed to watershed due to the number of OFRs, and the OFRs' different
purposes (e.g., different irrigation schedule) (Ayalew et al., 2017; Fowler et al., 2015; Habets et al.,
2018; Nathan & Lowe, 2012; Pinhati et al., 2020; Rabelo et al., 2021). As pointed out by Habets et
al., (2018) the mean annual decrease in flow from all studies was -13.4% ± 8%. Our results are
aligned with this value, which varied between -24.2 ± 4% and -14.6 ± 3% for all flow classes. In
addition, OFRs can reduce peak flow on average by 45% (Habets et al., 2018; Nathan and Lowe,
2012; Thompson, 2012), and up to 70% (Ayalew et al., 2017) for certain flow events. Likewise, our
results are consistent with these findings, in which the mean impact on peak flow varied
between -60.7 ± 12% and -43.9 ± 12%. Furthermore, differently from previous research, our
results showed that the OFRs may have a positive (< 9%) impact on flow (Fig. 5, classes 3 and
4). This could be explained by the level of details in our analyses. While we calculated the
monthly impact on flow at the channel scale by aggregating the OFRs to the closest channel,
previous studies have mostly reported the annual impact on flows (Habets et al., 2018), and
they performed their analysis at the subwatershed scale by aggregating the OFRs to a single
point at the outlet of each subwatershed in SWAT (Evenson et al., 2018; Kim & Parajuli, 2014;
Perrin, 2012; Zhang et al., 2012), or they used different modeling approaches (see Habet et al.,

(2018)).

By leveraging the latest improvements in SWAT+ to simulate water impoundments
(Molina-Navarro et al., 2018) in combination with a novel algorithm to monitor OFRs (Perin et





al., 2022), we modeled the impact of the OFRs on flow at the channel scale. In addition, the
surface area scenarios enabled us to account for events when the OFRs were at the lowest,
regular, and fullest capacities according to their surface area (see Fig. 2). This is an
improvement over previous studies (e.g., Ni et al., 2020; Ni and Parajuli, 2018; Perrin, 2012) that
used a single surface area (i.e., one snapshot in time) to represent the OFRs in SWAT. The small
differences (< 5%) between the surface area scenarios in terms of mean percent change on
monthly flow indicates that the OFRs' surface area variation had a low impact on flow. For
instance, during January and May the mean monthly percent change ranged between -35.8 ±
6% and -32.0 ± 7%, and during June and December it varied between -8.8 ± 5% and -5.4 ± 6%
for the three surface area scenarios. The same was observed for peak flow, with a mean
monthly impact ranging between -52.7 ± 17% and -49.4 ± 18%. This small variability on flow
impact was observed even though the total OFR surface area increased by 590 ha and 1194 ha
when comparing the lower scenario with the mean and upper scenarios (Fig. 5). However, the
OFRs represented a small portion (< 1%) of the total area of the modeled watershed (Fig. 1).
These findings are related to the fact that flow simulations are governed by several
hydrological processes (e.g., run-off, precipitation, evapotranspiration) besides the presence of
OFRs on the channel (Bieger et al., 2017; Dile et al., 2022; Arnold et al., 2012). In addition, when
assessing the percent change in flow at the channel scale, the differences in surface area
between the scenarios occurred at a lower magnitude when compared to the total OFRs
surface area. For instance, an OFR with surface area smaller than 10 ha, and with surface area
variations between 10 and 20% for the three scenarios, may not lead to differences (e.g., > 10%)
between the three scenarios.
4.2 OFRs impacts on flow and peak flow

Our findings highlight that the impacts of the OFRs on flow and peak flow have a

significant intra- and inter-annual variability (Figs. 5, 6, and 7), and the impacts vary according
to different OFRs and channels (Fig. 5). The largest impacts on flow occurred during the first





part of the year between January and May, the period of the year when the peak flows occur.
In addition, this time of the year also coincides with the period when the region receives most
of its precipitation (Perin et al., 2021b), and the OFRs are at their fullest capacity (i.e., OFRs
storing their maximum amount of water) (Perin et al., 2022). During the second part of the year,
we observed a milder mean percent change in flow for all flow classes and all scenarios, and a
greater variability in percent change, notably for the months of July and August (Fig. 5).
Moreover, most of the irrigation activities happen between June and September (Perin et al.,
2021b, Yaeger et al., 2017), and it is when the OFRs are at their lowest capacities (i.e., storing less
water) (Perin et al., 2022), which could explain their moderate impact and higher variability
during these months—even though we are not accounting for the OFRs inflows and outflows,
and not simulating irrigation events. Additionally, the variability of the OFRs impacts is related
to the OFRs' physical properties (e.g., surface area and location in the watershed). For example,
the OFR surface area will have an impact on flow and peak flow, as shown by the different
surface area scenarios, and depending on where the OFR is located in the watershed, given
that it may be connected to lower or higher flow channels, which contributes to their impact
variability during the year (Figs. 4 and 5). Besides the OFRs' physical properties, the built-in
complexity of SWAT—when simulating the presence of the OFRs and the various hydrological
processes (e.g., run-off, precipitation, evapotranspiration) governing the water cycle—
contributes to the differences in the OFRs impacts. This complexity is illustrated in Fig. 8
showing that the upper scenario can have a higher or lower impact on flow when compared
to the lower and mean scenarios.

When assessing the annual impact of the OFRs accounting for each subwatershed flow

compared to the main model outlet flow, and each subwatershed reservoir capacity (Fig. 9),
we found that even though the presence of the OFRs can have a significant impact on flow
(Figs. 5, 6, and 7), the highest impacts tend to occur on the subwatersheds that contribute the
least (< 10%) to the main model outlet. In general, the highest impacts occurred on the





channels with smaller flow magnitudes, and the subwatersheds with the highest reservoir
capacities did not have the highest impact on flow.
4.3 Research implications and applications to other study regions

Overall, we presented a new approach to quantitatively analyze the impact of a network

of OFRs on mean and peak flow, and we described the various potential reasons behind the
variability of the impacts. Our results indicate that OFRs do not have an equally distributed
impact on mean and peak flow across the watershed. Hence, assessing the OFRs location as
well as their numbers across the watershed is important when aiming to manage the
construction of new OFRs. In particular, the geospatial variability of the OFRs impacts could be
taken into account by water agencies when planning and developing a network of OFRs, given
it is possible to identify the areas that are under high pressure (e.g., regions with multiple OFRs
that are having a significant impact on flow), and to identify areas that could benefit from the
construction of new OFRs, targeting improvements on water resources management and
irrigation activities.

Furthermore, even though the OFRs impacts may vary significantly in different

watersheds (Habets et al., 2018), our approach could be transferable to other places across the
world, as it integrates a top-drown data-driven remote sensing-based algorithm, which is
based on freely available and private Earth Observations datasets, with the latest SWAT+
hydrological modeling developments. In addition, the widespread use of SWAT+ and its open-
source nature, is yet another factor contributing to the transferability of the novel approach
presented in this study. This is relevant as the number of OFRs is expected to increase globally
(Althoff et al., 2020; Habets et al., 2014; Habets et al., 2018; Krol et al., 2011; Rodrigues et al., 2012),
with a limited knowledge of how the OFRs may impact surface hydrology in different
watersheds, and under diverse environmental conditions. Finally, in tandem with the OFRs'
key role on irrigated food production, in part to adapt to climate change (Habets et al., 2018)
and to alleviate the pressure on surface and groundwater resources (Vanthof & Kelly, 2019;



Yaeger et al., 2017; Yaeger et al., 2018), their impacts on surface hydrology need to be
considered to avoid exacerbating the surface water stress already intensified by climate
change and population growth (Vörösmarty et al., 2010).
## 5 Future improvements
Future improvements should focus on how to better represent the OFRs water management
(i.e., OFRs inflows and outflows) in SWAT+. Given that each OFR has an independent water
balance, accounting for the OFRs water volume change would be a more realistic
representation of the OFRs when compared to the three surface area scenarios tested in this
study. Estimating the OFRs volume change can be done by combining the OFRs surface area
time series with area-elevation equations—these equations describe the OFRs' bathymetry,
and allow volume estimation by inputting the OFRs' surface area (Liebe et al., 2005; Meigh,
1995; Sawunyama et al., 2006). After carefully assessing different methods to derive these
equations (Arvor et al., 2018; Avisse et al., 2017; Li et al., 2021; Meigh, 1995; Sawunyama et al., 2006;
Vanthof & Kelly, 2019; Yao et al., 2018; Zhang et al., 2016), we decided that measured ground-
data of the OFRs' depth—which is not available—is required to estimate the equations with an
acceptable uncertainty. Estimating the area-elevation equations entails several challenges,
including: 1) despite the fact that there are several DEMs available for the study region
(Arkansas GIS Office, 2022)—DEMs can be used to estimate the OFRs bottom elevation—the
DEMs were collected when most of the OFRs were full (i.e., bathymetry was not exposed),
which limits their use in this case; and 2) although the OFRs are located within the same
geomorphological region, they have different depth, shape and physical characteristics (Perin
et al., 2022; Yaeger et al., 2017). Therefore, even if a generalized area-elevation equation was
calculated for our study region—this is a common approach done by other studies (Mady et
al., 2020; Vanthof and Kelly, 2019)—that would still lead to high uncertainties of water volume
changes. Ideally, each OFR would have its own equation, which was not possible when this
study was done.


Efforts should also be made to improve SWAT+ capabilities to receive measured OFRs'
inflows and outflows. The latest version of the model has improved the hydrological
representation of small water impoundments in SWAT+ (Mollina-Navarro et al., 2018).
Nonetheless, at the time of our study, the newest version of the model does not allow users to
input measured or calculated OFRs' inflows and outflows. Instead, the model developers
recommend simulating the OFRs water balance using decision tables (Arnold et al., 2018; Dile
et al., 2022). However, there are very limited guidelines on how to create these decision tables.
In addition, the tables would simulate the OFRs water balance instead of using the measured
or calculated volume change, which could introduce more uncertainties to the modeling
scenarios.
6 Conclusions
We proposed a novel approach that combines a top-down data driven remote sensing-based
algorithm with the latest developments in SWAT+ to simulate the cumulative impacts of OFRs.
This enabled us to assess the spatial and temporal variability of the OFRs impacts, as well as
the intra- and inter-annual impact changes on mean and peak flow, at the watershed and
subwatershed levels. Incorporating Earth Observation derived information with a hydrological
model, allowed us to capture the dynamic changes of the OFRs, and to simulate their impacts
under different OFR capacity scenarios.
Our study showed that the OFRs may have an impact on flow and peak flow, which can
have a significant inter- and intra-annual variability. The impact of the OFRs is not equally
distributed across the watershed, and it varies according to the OFRs spatial distribution, and
their surface area (i.e., water storage capacity). As the number of OFRs is expected to increase
globally—partially to adapt to climate change and to alleviate pressure on groundwater
resources—and therefore, also increase their relevance to irrigated food production, it is
imperative to develop new frameworks to further understand the OFRs impacts on surface
hydrology. In this regard, we provided a combination of different methods that can be used in





other watersheds, which can support water agencies with information to improve surface
water resources management.

## 7 Author contribution

VP, MGT planned study, analyzed data and modeling, and wrote and reviewed the
manuscript. SF and AS carried out software analyses, wrote and reviewed. MLR and MAY data
curation, wrote and reviewed.

## 8 Competing interests

The contact author has declared that none of the authors has any competing interests.

## 9 Acknowledgments

The first author was supported by NASA through the Future Investigators in NASA Earth and
Space Science and Technology fellowship.

## 10 Data Availability

The Soil Water Assessment Tool (SWAT) hydrological model and all necessary tools to
perform calibration, validation, and data analyses can be accessed through SWAT's online
portal: https://swat.tamu.edu/.
The National Land Cover Database (30 m) (Homer et al., 2020) and the Gridded Soil Survey
Geographic Database (gSSURGO) (Soil Survey Staff, USDA-NRCS, 2021) (100 m) are accessible
through the USGS's portal: https://www.usgs.gov/centers/eros/science/national-land-cover-
database, and here https://www.nrcs.usda.gov/resources/data-and-reports/gridded-soil-
survey-geographic-gssurgo-database, respectively.
The climate data extracted from the Gridded Surface Meteorological Datasets (Abatzoglou,
2013) is available in Google Earth Engine (Gorelick et al., 2017), here
https://developers.google.com/earth-engine/datasets/catalog/IDAHO_EPSCOR_GRIDMET.
The Kalman filter derived surface area time series is available through Perin et al., (2022).

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
