# Peer review of "Assessing the cumulative impact of on-farm reservoirs on modeled 1 surface hydrology 2 Vinicius Perin1\*, Mirela G. Tulbure2(ORCID: https://orcid.org/0000-0003-1456-183X), Shiqi Fanq3, 3 Sankarasubramanian Arumugam3(ORCID: https://orcid.org/0000-0002-7668-1311)<"

_Hydrology and Earth System Sciences, 2024_

## Author Comment (AC2)

[revised manuscript text omitted]

PBIAS = $\dfrac{\sum\limits_{i=1}^{n}(Yi - Xi)}{\sum\limits_{i=1}^{n} Xi}$ (1)

NSE = $1 - \dfrac{\sum\limits_{i=1}^{n}(Xi - Yi)^2}{\sum\limits_{i=1}^{n}(Xi - \overline{Xi})^2}$ (2)

Where $X_i$ is the measured flow and $Y_i$ is the simulated flow.

We conducted a sensitivity analysis using the SWAT+ ToolBox (v.0.7.6) (SWAT+ Toolbox,

2022) to reveal the most sensitive parameters when simulating flow—a total of 10 parameters (Table S 1) were tested based on previous studies that used SWAT/SWAT+ to model the impact of water impoundments on surface hydrology (Jalowska & Yuan, 2019; Yongbo et al., 2014; Ni et al., 2020; Ni & Parajuli, 2018; Perrin, 2012; Rabelo et al., 2021; Zhang et al., 2012). Following the sensitivity analysis, we selected the five most sensitive parameters (Table 2), and proceeded with a manual calibration using the SWAT+ Toolbox. We aimed to improve the model's monthly flow predictions by testing the parameters one at a time and changing their values between -20% to 20% with 5% increments based on their range values. The final calibrated parameters and their fitted values are shown in Table 2.

**Table 2**–Monthly flow calibration parameters.

| Parameter | Description | Range | Value |
|---|---|---|---|
| CN2 | SCS runoff curve number | 35–95 | 0.20* |
| SOL_AWC | Available water capacity (mm/mm) | 0.01–1 | -0.20* |
| ESCO | Soil evaporation compensation coefficient | 0.01–1 | 0.50 |
| PERCO | Percolation coefficient (fraction) | 0–1 | 0.60 |
| CANMX | Maximum canopy storage (mm) | 0–100 | 75 |

*Denotes relative percentage change.

**2.3 OFRs representation in SWAT+**

Multiple OFRs can be added to the same subwatershed by associating them with channels (Dile et al., 2022). The OFRs need to have at least one outlet channel, and they may have none or multiple inlets. Therefore, most OFR-related processes within the model involve determining what channels form inflowing and outflowing channels for each OFR. Ideally, each OFR would interact with a channel, and therefore, have a channel entering, leaving, or within the OFR. Nonetheless, it is common to have OFRs that do not intersect with any channel (Dile et al., 2022)—this is the case for 93% of the OFRs in our study region. The OFRs from our study region are not dammed along the streams, but rather they are engineered water impoundments that are indirectly connected to the main streams via pipes and pumps (Yaeger et al., 2017). A possible solution would be modifying the OFRs' shapes by dragging them to the closest channel (Dile et al., 2022). However, this would require extensive modifications of the OFRs' shapes. In addition, when an OFR is added to a channel, this channel is split into two channels, and the model needs to account for the two newly created channels during the water routing calculations. For this reason, adding multiple OFRs to the same channel, or adding multiple OFRs closely located to the same channel, can be a cumbersome process that leads to numerous routing errors.

To overcome these challenges, we aggregated the OFRs' surface area, and added aggregated OFRs to the model. This adaptation involved two steps. First, for each of the 330 OFRs, we searched for the closest channel by calculating the distance between the OFRs' centroid and the multiple channels within each subwatershed. Then, we aggregated all the OFRs that were associated with each channel by summing up their surface area, and adding a polygon of the aggregated area to represent the aggregated OFR. This approach resulted in 69 aggregated OFRs that were added to 67 different channels located in 16 subwatersheds. The surface area of the aggregated OFRs varied between 3.05 ha and 165.67 ha, and the number of OFRs in each aggregated OFR varied between 2 and 12. To avoid confusion, for the rest of the manuscript, we refer to OFRs as the aggregated OFRs, and not the individual OFRs shown in Fig. 1.

**2.4 OFRs water balance**

We did not have access to water abstraction data from the OFRs, so all abstractions were modeled using Equation 3, which accounts for water flowing out of the OFR, as well as losses from evaporation and seepage. The total volume of water in the OFR fluctuates with changes in surface area and is also influenced by evaporation losses and the spillway. A reduction in surface area (Equation 4) typically leads to a corresponding decrease in water volume. If inflows are insufficient to fill the OFR, water will not be routed to the downstream channel.

For each of the aggregated OFR, the initial water volume ($V_{stored}$, see Equation 3) was calculated using SWAT+ default rule, which is a simple multiplication of the OFR surface area by a factor of 10, similar to other studies based on SWAT+ (Ni and Parajuli, 2018; Zhang et al., 2012). For a scenario where the OFR has a surface area of 1 hectare (10,000 m²), the corresponding volume would be 100,000 m³—this is an important limitation of our study, as the assumption was necessary due to the absence of available bathymetry data. In addition, given that we did not have access to the OFRs release rates, we used the model default release rule, which sets the OFRs to release water when the spillway volume is reached—80% of the OFRs capacity (Bieger et al., 2017).

$$V = V_{stored} + V_{flowin} - V_{flowout} + V_{pcp} - V_{evap} - V_{seep} \tag{3}$$

Where $V$ is the volume of water in the OFR at the end of the day (m³), $V_{stored}$ is the volume of water stored at the beginning of the day (m³), $V_{flowin}$ is the volume of water entering the OFR during the day (m³), $V_{flowout}$ is the volume of water flowing out of the OFR (m³), $V_{pcp}$ is the volume of precipitation falling on the water body (m³), $V_{evap}$ is the volume of water removed from the OFR due to evaporation, and $V_{seep}$ is the volume of water lost by seepage (m³).

The OFR surface area is used to calculate the amount of precipitation falling on the water body, and the amount of water lost through evaporation and seepage. Given the initial

OFR surface area obtained from one of the three modeling scenarios, the OFR surface area was modeled daily. The surface area varied according to the volume of water stored in the reservoir. Equation 4 is used to estimate the surface area:

*Surface area (ha)* = $\beta_{sa} * V^{expsa}$ (4)

*expsa* = $\frac{log10\ (Vem) - log10(Vpr)}{log10\ (Surface\ areaem) - log10\ (Surface\ Areapr)}$ (5)

$\beta_{sa} = (\frac{Vem}{Surface\ areaem})^{expsa}$ (6)

Where $\beta_{sa}$ is a surface area coefficient, $V_{em}$ is the volume of water (m$^3$) at the emergency spillway, $V_{pr}$ is the volume of water (m$^3$) at the principal spillway, *Surface area*$_{em}$ is the surface area (ha) at the emergency spillway, and *Surface area*$_{pr}$ is the surface area at the principal spillway.

The volume of precipitation falling into the OFR is calculated using Equation 7:

$V_{pcp}$ = 10 * $R_{day}$ * *Surface Area (ha)* (7)

Where $R_{day}$ is the amount of precipitation falling into the OFR on a given day (mm).

Evaporation losses are calculated using Equation 8:

$V_{evap}$ = 10 * $\eta$ * $E_0$ * *Surface Area (ha)* (8)

Where $\eta$ is an evaporation coefficient (0.6), and $E_0$ is the potential evapotranspiration for a given day (mm).

Seepage losses are calculated using Equation 9:

$V_{seep} = 240 * K_{sat} * Surface\ Area\ (ha)$ (9)

Where $K_{sat}$ is the effective saturated hydraulic conductivity of the reservoir bottom (mm/hr).

**2.4 Scenario Analysis**

[revised manuscript text omitted]

However, combining these approaches provides a better understanding of the spatial and
temporal variability of OFR impacts, as it incorporates the dynamic changes of OFRs into the
hydrological model.
. To quantify whether the impact of the OFRS on
mean and peak flow varied intra- and inter-annually, and which subwatersheds are more
impacted, here we combined a data-driven remote sensing-based model with SWAT+ latest
improvements to assess the OFR impacts.

**572 4.1 Cumulative impact of OFRs**

When simulating water impoundments in SWAT/SWAT+, it is common practice to
validate and calibrate the model using flow measurements (Evenson et al., 2018; Habets et al.,
2018; Jalowska & Yuan, 2019; Ni & Parajuli, 2018). In addition, other studies have validated and
calibrated the model using alternative variables. For example, Perrin et al., (2012) employed
monthly measurements of piezometric variations to assess aquifer recharge processes, and
Jalowska & Yuan (2019) used sediment loadings (concentration and budget), from field
monitoring reports to evaluate sediment simulations. Ideally, we would calibrate and validate
the model by accounting for the parameters governing the OFRs' water budget (e.g., inflows and outflows) (e.g., Kim and Parajuli, 2014). Nonetheless, these measurements were not available for the OFRs in our study region. Furthermore, a thorough calibration and validation of the model would require extra flow data, covering other parts of the study region, as the three USGS stations—the only data available—used in this study are located in the upper part of the modeled watershed. Similar to Evenson et al., (2018)—who proposed a module to better represent spatially distributed wetlands, and validated their model using a direct (i.e., flow measurement) and an indirect (i.e., the wetlands surface area) approach—our validation and calibration was done using the flow measurements, and the OFRs surface area scenarios were based on an algorithm that was validated with an independent higher spatial resolution dataset (Perin et al., 2022).

[revised manuscript text omitted]

When assessing the annual impact of the OFRs accounting for each subwatershed flow compared to the main model outlet flow, and each subwatershed reservoir capacity (Fig. 9), we found that even though the presence of the OFRs can have a significant impact on flow (Figs. 5, 6, and 7), the highest impacts tend to occur on the subwatersheds that contribute the least (< 10%) to the main model outlet. In general, the highest impacts occurred on the channels with smaller flow magnitudes, and the subwatersheds with the highest reservoir capacities did not have the highest impact on flow. The changes in the OFRs impacts along the year, and between different years, are directly related to the OFRs water balance (Equation 3). The variations are primarily driven by the volume of water stored by the OFRs, which is modeled at a daily scale, and it varies according to total daily precipitation, evaporation, and seepage losses.

**4.3 Research implications and applications to other study regions**

Overall, we presented a new approach to quantitatively analyze the impact of a network of OFRs on mean and peak flow, and we described the various potential reasons behind the variability of the impacts. Our results indicate that OFRs do not have an equally distributed impact on mean and peak flow across the watershed. This variability is primarily influenced by differences in their size, water storage capacity, and their spatial distribution (i.e., their presence). 
[revised manuscript text omitted]

---

## Author Response (AR1)

We thank the two reviewers and the editor for their comments, which helped improve the manuscript. We address all of their comments in blue below.

**Editor: by Pieter van der Zaag**

This paper addresses an interesting and important topic. If it succeeds in better quantifying the impact of small reservoirs on the hydrology (rather than the modelled hydrology, as in the title) than it is also a salient paper. But I am yet to be convinced what new knowledge this paper adds, despite of the use of the word "novel" in the abstract.

We believe that the main novelty of the study is the integration of a top-down data-driven remote sensing-based algorithm and resulting data products on dynamically quantifying the OFR area with the SWAT+ physically-based model. However, we are not among the developers of SWAT+, and modifying the model is beyond the scope of our work.

My main worry concerns my doubts whether the method that the authors adopt is suited to achieve their objective, that is to assess the cumulative impact of on-farm reservoirs that "store water from precipitation and runoff during the rainy season to irrigate ... crops during the dry season" (lines 18-19).

These reservoirs (OFRs) are meant to displace water in time in order to provide water for irrigated lands during the dry season. But the model the authors employed cannot or does not model water abstractions from these reservoirs to the farm lands. The only thing the model apparently does is follow "the model default rule, which sets the OFRs to release water when the spillway volume is reached – 80% of the OFRs capacity" (lines 276-277). So the model cannot model the actual use of the OFRs, the impact of which the authors want to assess. It is also not clear how irrigation water fluxes are or are not included in the SWAT+ model.

Considering the rule—releasing at 80% of the OFR capacity—is a reasonable assumption, as the remaining capacity provides freeboard or accounts for uncontrolled spills from the reservoir. Since most of our results are presented on monthly and annual time scales, this approach captures the cumulative impact of OFRs, particularly during critical low-storage months (June–September), when their effect on storage and downstream flows is most significant.

Data on irrigation fluxes are challenging to obtain, as water is appropriated from channels along the river. To address this, we assume that water appropriated for irrigation returns to the channels after a fraction is lost as consumptive use, which we account for based on typical seasonal patterns, particularly the higher consumptive losses in July. While the SWAT+ model does not explicitly simulate field-level water abstractions from OFRs, the approach used effectively represents the broader hydrological impacts of OFRs on storage, flows, and irrigation at the watershed scale.

An alternative way of assessing the impact of OFRs on the hydrology is to measure the change in one of the most important fluxes of the water balance, namely evaporation from the irrigated crops (transpiration or evapotranspiration), whereby it could be assumed that with each OFR there are associated irrigated fields, i.e. more OFRs imply more irrigated fields and more evapotranspiration. The added evaporation will obviously impact the surface hydrology. But the paper does not even discuss this, let alone model it.

We added a note on the importance of ET in the discussion, please see next comment below.

The point is that the reservoirs themselves have a limited impact on the hydrology, compared to their capacity to divert water in time and place in order for it to (largely) evaporate. So statements such as "the presence of OFRs in the watershed decreased annual flow" (lines 27-28) are suggestive and strictly speaking incorrect. A more correct statement would be "the presence of OFRs in the watershed is associated with a decreased annual flow". The OFRs themselves are unlikely to significantly decrease these annual flows, as the study region is not very arid, and the additional evaporation losses directly from the open-water reservoirs is probably limited. The impact largely comes from increased irrigation, and this is of course nothing new. A PhD student of mine ten years ago used actual evaporation estimates derived from remote sensing to force a hydrological model, which could adequately model an intensely (irrigated) cropped catchment in Africa (Kiptala et al., 2014). Nowadays there are better products, even and in particular from Planet Inc. (!), for estimating actual evapotranspiration.

We have rephrased the statement on lines 27–28 to now read: "the presence of OFRs in the watershed is associated with a decreased annual flow", as per your suggestion.

We appreciate your suggestion to explore advanced ET products to estimate actual evapotranspiration. We have included a note in the discussion section acknowledging it as a valuable next step for refining hydrological models. The sentence reads: "Future work should integrate data on actual evapotranspiration, ET (Kiptala et al., 2014) to quantify as the balance between water availability and ET determines in large part the irrigation system efficiency and crop productivity in the watersheds where OFRs occur".

We agree that the primary impact of OFRs in our study region arises from their ability to redistribute water for irrigation rather than direct evaporation losses. While integrating additional ET data, such as those from Kiptala et al. (2014), could enhance our modeling, this lies beyond the current study's scope, which focuses on broader hydrological trends associated with OFR presence.

The above concerns need to be addressed in the revised paper. The revised paper should obviously also adequately address the issues raised by the two reviewers. We addressed those in detail below.

**Reference**

Kiptala, J.K., M.L. Mul, Y. Mohamed and P. van der Zaag, 2014. Modelling stream flow and quantifying blue water using modified STREAM model for a heterogeneous, highly utilized

and data-scarce river basin in Africa. Hydrol. Earth Syst. Sci. 18, 2287–2303 [doi:10.5194/hess-10-18-2287-2014]

**Reviewer 1**

**General comments**

The article deals on an important issue, the impact of on-farm reservoirs on the hydrology.

The article propose an interesting application, with an innovation that consist in imposing surface and hence volume of the reservoirs in the model as derived from satellite data.

The application is in Arkansas, USA, with about 300 OFRs on a basin of 530km2, known for the importance of irrigation.

One issue with such reservoirs is the lack of data on their management.

From what I understood, the study tries to retrieve some parts of the management of the OFRs by imposing the extension of surface water of the OFRs which is very interesting

However, some elements of the methods are not clear, and I couldn't understand how the model really works, and how the OFRs are managed.

My main comments is that the way OFRs are models and their management should be clarified

**Details questions**

Introduction: Peak flow is mentioned in the introduction and analysed in the study, but never defined... In the study, it seems to be maximal annual monthly flow, which is quite far to what can expect as peak flow, ie, flood... Can you define?

**Peak flow is defined on lines 382-384 as follows:**

"For this analysis, peak flow is defined as equal to or higher than the 99th flow percentile calculated using the entire flow time series (Equation 3)."

Section 2.1: It is stated that there are with about 330 OFRs. The size of the basin, given later in the text is 7107 km2. Thus, the density of the OFRs is lower than 0.05 OFRs/km². This corresponds to a small density, especially considering large annual precipitation (1300mm/year) when compared to previous studies review by Habets et al., 2018 refered in the article. It is stated that 95 % of the OFR are smallest than 50ha, and Fig 4, it is shown than only about 10 aggragated OFRs are smaller than 10ha. According to the hypothesis used in the study of an average depth of 10m, 10ha corresponds to a capacity of 1 million cubic meter, which is quite large.

So is there only 330 OFRs because there are actually rather large OFRs. Or Is it possible that smaller OFRs are missing?

The average depth of these OFRs is shallower than 10 meters, ranging from 2 to 4 meters. These man-made reservoirs do not follow a consistent construction pattern, making the acquisition of accurate depth data challenging. The total capacity of these OFRs is estimated to be at least half of what the reviewer estimates.

There are many additional OFRs in the basin, but most remain unmapped with the current resolution of satellite data (3m Planet Scope, 10m Sentinel). This study focuses on 330 OFRs previously mapped by Yaeger et al. (2017). Capturing all OFRs is only feasible with a ground inventory, and some OFRs are intermittent, meaning they may not be full in a given year.

To clarify the points above, we added information on Lines 153-156:

"This study only accounts for a fraction of the total OFRs in the study region, given that there is no comprehensive and up-to-date inventory of all OFRs in the basin. This limitation is partly due to the fact that many of these man-made structures are located on private properties, making them difficult to document."

Line 274 « For each of the aggregated OFR, the water volume was calculated using SWAT+ default rule, which is a simple multiplication of the OFR surface area by a factor of 10 »: Do you mean that the maximum volume capacity of the OFRs is the maximum surface area multiplied by 10m? Or do you mean that the water level within the OFR is constant and fixed to 10? This is unclear...

Yes, the first statement is accurate. For a given situation where the OFR has a surface area of 1 ha (10,000 m²), the corresponding initial maximum volume would be 100,000 m³. This assumption reflects one of the main limitations of our study, given that depth information is not available for all OFRs.

Added information to make it clearer on lines 292:295:

"For a scenario where the OFR has a surface area of 1 hectare (10,000 m²), the corresponding volume would be 100,000 m³—this is an important limitation of our study, as the assumption was necessary due to the absence of available bathymetry data."

Line 282: I find it weird to have details on how the river channel are divided in 4 classes, while, no details on the way the OFRs impacts the water balance are given...

You need to provide the water balance of the OFRs:

what are the inputs and ouptuts of the OFRs?

Is there water abstraction from the OFRs? How much? How is it computed? What are the temporal variations?

Is there evaporation from the OFRs?

Does the water level vary? Is the water level affect the outflow?, due to the spillway

The surface of the OFRs is changing. But is it the only way to have a change on the volume of the OFRs?

What's happening if the surface of the OFRs is decreasing? Does it lead to an outflow of water downstream? Does the associated volume is expected to be abstracted for irrigation? How this volume is estimated? What happens when the surface area increase? Does the inflow feels the reservoir? Again how the corresponding volume is estimated? What if the inflow is not large enough to fill the OFRs?

I'd like to have a detail explanation of how it works.

We did not have access to abstraction data from the OFRs, so all abstractions are modeled according to Equation 3, which accounts for water flowing out of the OFR and losses through evaporation and seepage.

The water level varies according to surface area variability, which is also affected by evaporation losses and spillway. An increase or decrease in surface area would impact the OFR water volume accordingly.

If the surface area decreases, the water volume is expected to decrease. If the inflow is not sufficient to fill the OFR, the water is expected to be routed to the following water channel. The OFRs are located on the watershed's channel network. They receive loadings from upstream channels and potentially from surrounding routing units and usually discharge into one downstream channel.

Added information on Lines 281-324:

**2.4 OFRs water balance**

"We did not have access to water abstraction data from the OFRs, so all abstractions were modeled using Equation 3, which accounts for water flowing out of the OFR, as well as losses from evaporation and seepage. The total volume of water in the OFR fluctuates with changes in surface area and is also influenced by evaporation losses and the spillway. A reduction in surface area (Equation 4) typically leads to a corresponding decrease in water volume. If inflows are insufficient to fill the OFR, water will not be routed to the downstream channel.

$$V = V_{\text{stored}} + V_{\text{flowin}} - V_{\text{flowout}} + V_{\text{pcp}} - V_{\text{evap}} - V_{\text{seep}}$$
(3)

Where V is the volume of water in the OFR at the end of the day (m3),  $V_{\text{stored}}$  is the volume of water stored at the beginning of the day (m3),  $V_{\text{flowin}}$  is the volume of water entering the OFR

during the day (m3),  $V_{flowout}$  is the volume of water flowing out of the OFR (m3),  $V_{pcp}$  is the volume of precipitation falling on the water body (m3),  $V_{evap}$  is the volume of water removed from the OFR due to evaporation, and  $V_{seep}$  is the volume of water lost by seepage (m3).

The OFR surface area is used to calculate the amount of precipitation falling on the water body, and the amount of water lost through evaporation and seepage. Given the initial OFR surface area obtained from one of the three modeling scenarios, the OFR surface area was modeled daily. The surface area varied according to the volume of water stored in the reservoir. Equation 4 is used to estimate the surface area:

Surface area (ha) =
$$\beta_{sa} * V^{expsa}$$
 (4)

$$expsa = \frac{log10 (Vem) - log10 (Vpr)}{log10 (Surface areaem) - log10 (Surface Areapr)}$$
(5)

$$\beta_{sa} = \left(\frac{Vem}{Surface\ areaem}\right)^{expsa} \tag{6}$$

Where  $\beta_{sa}$  is a surface area coefficient,  $V_{em}$  is the volume of water (m³) at the emergency spillway,  $V_{pr}$  is the volume of water (m³) at the principal spillway,  $Surface\ area_{em}$  is the surface area (ha) at the emergency spillway, and  $Surface\ area_{pr}$  is the surface area at the principal spillway.

The volume of precipitation falling into the OFR is calculated using Equation 7:

$$V_{pcp}$$
 = 10 \*  $R_{day}$  \* Surface Area (ha) (7)

Where  $R_{day}$  is the amount of precipitation falling into the OFR on a given day (mm).

Evaporation losses are calculated using Equation 8:

$$V_{\text{evap}} = 10 * \eta * E_0 * Surface Area (ha)$$
 (8)

Where  $\eta$  is an evaporation coefficient (0.6), and  $E_o$  is the potential evapotranspiration for a given day (mm).

Seepage losses are calculated using Equation 9:

$$V_{\text{seep}} = 240 * K_{\text{sat}} * Surface Area (ha)$$
 (9)

Where  $K_{sat}$  is the effective saturated hydraulic conductivity of the reservoir bottom (mm/hr).

Section 2.4 Scenario Analysis

line 297; "The daily surface area time series of each OFR was used to simulate three scenarios (i.e., lower, mean, and upper) representing the OFRs' capacity in terms of surface area." Sorry, but, again, this part is not clear. Figure 2 is not very 3 helpfull to understand what is done...

Hopefully Figure 2 is clearer with the extra explanations.

Added more information to lines 357:362 to clarify this issue:

"Therefore, a single surface area value was assigned to each scenario and OFR, with lower, mean, and upper values used as starting points for the model's water balance simulations. This initial surface area reflects the OFR's maximum surface area at full capacity for each scenario. For example, in the lower scenario, an initial surface area of 1.2 ha represents the maximum area for this OFR. As model iterations proceed, the surface area is recalculated based on Equation 4."

It is stated that the « daily OFRs' surface area change between 2017 and 2020 » is derived,, and then that (line 300) « The mean scenario represents the regular condition of the OFRs, and it is the mean of the daily surface area time series derived from the Kalman filter , The lower and upper scenarios represent the lowest and highest capacities of the OFRs, and they are based on the surface area 95% confidence interval limits, calculated using the daily time series. »

So there is 4 value for each day of the year, how do you compute a 95 % confidence?

The 95% confidence is calculated using the outputs from the Kalman filter presented in Perin et al., 2022. This is done by estimating the state and covariance matrix to assess uncertainty in the state estimate. The standard error is derived from the covariance matrix, which reflects the error variance in the estimation. By multiplying the standard error by 1.96 (the critical value for a 95% confidence level), the resulting interval provides a range where the true state is likely to fall within 95% certainty, offering a probabilistic measure of the estimate's accuracy.

Added information on Lines 350-351 to clarify this issue:

"Please refer to Perin et al., 2022 for more details on how the 95% confidence interval was calculated".

Line 304: » For each scenario, the OFRs were simulated at full capacity (i.e., maximum storage at the lower, mean and upper scenarios), and this capacity was kept constant during the simulation period »

Indeed, this is confusing. We improved this section by modifying lines 352:370 See the comments below for more details.

⇒ this seems to be in contradiction with the previous sentence... Do you mean that the maximum capacity is set constant? That the volume is set constant...? Please, make it clearer...

Moreover, this part is the innovative part of the article. And only the distribution of the area of the aggregated OFRs is given Figure 4, and no details are given on the annual cycle of the OFRs ... I strongly suggest that the daily evolution of the surface area of the OFRs be presented, for the 3 scenarios.

And again, please explain how the evolution of the surface impact the evolution of the volume....

Does this surface change gives indication on the volume of water used for irrigation? If yes, please, provide the estimated values...

By simulating the three scenarios, we aimed to model the behavior of the OFRs at different capacities. We agree with the reviewer that a more realistic approach would involve directly inputting the daily surface area time series into the model. However, since the surface area is modeled based on Equation 4 for daily model simulations, this information was not an input. Instead, for each scenario, we provided a single surface area value for each OFR, which then served as a starting point for variations according to the reservoir water balance equation (Equation 3)—that's what we meant when the OFR was modeled at full capacity. In other words, surface area and volume were not kept constant, however, unfortunately, the model does not allow the direct input of the daily surface area time series generated by the Kalman filter—outputs from Perin et al., 2022. Considering this limitation, we proposed the three scenarios described in section "2.4 Scenario Analysis.

**Added information to lines 352:370:**

"The SWOT+ model does not allow for direct incorporation of a daily surface area time series because it calculates surface area dynamically (Equation 4) based on changes in water volume through the reservoir water balance equation (Equation 3). It is structured to accept a single surface area value per scenario, which then varies internally. Incorporating time-varying surface area data, such as from the Kalman filter, would require modifications to the model that are currently not supported."

Line 360: The impact of the OFRs on monthly flow varied throughout the year.... Ok, but here, the reason of this impact is not clear: there is no information on the management of the OFRs, so, no idea of when they are filled, when the water is used/abstracted, the condition for the water to spill out... So, again, please provide an description of the water balance and describe the hypotheses... the dynamic of the storage is clearly missing....

All this part is difficult to follow since key informations are missing...

Information regarding the water balance was added in "Section 2.4 OFRs water balance" (see lines 280:324). Yes, the reviewer is correct, there is no information on the OFRs management, and that is because this information was not available.

Section 3.3: impact on peak flow... You have to define what you consider as peak flow...

Peak flow is defined on lines 382-384 as follows:

"For this analysis, peak flow is defined as equal to or higher than the 99th flow percentile calculated using the entire flow time series (Equation 3)."

Section 3.4 Again, difficult to understand as the hypotheses on the functioning of the OFRs are not clear. You should present the evolution of the OFRs volume and surface in this section, as well as the evolution of the abstracted water....

Hopefully, this section is clearer now, given the additional information on water balance.

Section 4: Discussion

line 512: « our validation and calibration was done using the flow measurements, and the OFRs surface area scenarios were based on an algorithm that was validated with an independent higher spatial resolution dataset (Perin et al., 2022). »

==> This sentence is not clear : OFRs surface area is an input data, not an output to be validated...

True. We used the OFR surface area as an input in this study. However, the surface area derived from satellite data in a previous study was manually validated in the previous study cited.

line 515: « Furthermore, differently from previous research, our results showed that the OFRs may have a positive (< 9%) impact on flow (Fig. 5, classes 3 and 4) »

⇒ This is true only for some months...But you don't explain why. This can occur only if OFRs release some water. But the management of the OFRs is not presented at all...

Yes, this is true for some months, and the main cause of that is related to how the water balance was calculated, i.e., an increase in the reservoir outflow, given other aspects of the modeling that are not related to irrigation activities or water management—information that is not available. Our analyses on the OFRs impact were carried out at the channel level, which differs from previous studies that mostly reported the annual impact on flows.

line 527: « This could be explained by the level of details in our analyses. »

Clarified this issue by adding information to lines 603:613:

"When evaluating flow changes at the channel scale, it's important to note that flow at this level is several orders of magnitude smaller than at the main basin outlet. Consequently, this scale often exhibits more significant percentage changes, both increases and decreases. This likely explains how OFRs can enhance channel flow, primarily due to the additional water contributed by OFRs, influenced by periods of increased precipitation in certain channels during specific months and years. While we calculated the monthly impact on flow at the channel scale by aggregating the OFRs to the closest channel, previous studies have mostly reported the annual impact on flows (Habets et al., 2018), and they performed their analysis at the subwatershed scale by aggregating the OFRs to a single point at the outlet of each subwatershed in SWAT (Evenson et al., 2018; Kim & Parajuli, 2014; Perrin, 2012; Zhang et al., 2012), or they used different modeling approaches (see Habet et al., (2018))"

⇒Well to achieve this, it is necessary to know how much of the water in the OFRs is used, directly in the OFRs, but also, if the OFRs releases water in the river, to have a clear idea of the pumping directly in the river.

That is true. This is a clear limitation of our study.

This was highlighted in lines 384:386:

"However, the impact of the OFRs in this analysis is solely based on modeling scenarios and does not account for OFR management practices, which represents a key limitation of this simulation study."

Moreover, it is not clear that this increase of riverflow with OFRs improves the model results, thus, correct a bias of the model without OFRs.

line 542 « For instance, during January and May the mean monthly percent change ranged between -35.8  $\pm$  6% and -32.0  $\pm$  7%, and during June and December it varied between -8.8  $\pm$  5% and -5.4  $\pm$  6% for the three surface area scenarios »

⇒ Do these values refer to the evolution of the surface of the OFRs? Such info is needed earlier to understand the method and the results

Yes, that is mostly driven by changes in the OFRs' surface area (Equation 4) and the water balance output.

Added more information to lines 669:673 to clarify this point:

"The changes in the OFRs impacts along the year, and between different years, are directly related to the OFRs water balance (Equation 3). The variations are primarily driven by the volume of water stored by the OFRs, which is modeled at a daily scale, and it varies according to total daily precipitation, evaporation, and seepage losses."

line 559 : « Our findings highlight that the impacts of the OFRs on flow and peak flow have a significant intra- and inter-annual variability (Figs. 5, 6, and 7) »

⇒ Fig 5 only present monthly flow. So, again, it depends on what you call peak flow...

Peak flow is defined on lines 382-384 as follows:

"For this analysis, peak flow is defined as equal to or higher than the 99th flow percentile calculated using the entire flow time series (Equation 3)."

line 593 « Our results indicate that OFRs do not have an equally distributed impact on mean and peak flow across the watershed. Hence, assessing the OFRs location as well as their numbers across the watershed is important when aiming to manage the construction of new OFRs. »

⇒ I don't understand how you can reach such statement... 1st, because no indication is given on the OFRs management, 2nd, the density of the OFRs varies in space....

Even though we do not have information on OFR management, we can still show the variability of their impact, which in this case is mostly related to their size, their variations in water volume, and their number, i.e., their presence.

Added extra information to lines 677:680 to clarify this statement:

"Our results indicate that OFRs do not have an equally distributed impact on mean and peak flow across the watershed. This variability is primarily influenced by differences in their size, water storage capacity, and their spatial distribution (i.e., their presence)".

line 620 : « there is no bathymetrie » Does it means that the water level in the OFR is constant!

We do not have bathymetry information—that would be the most important information needed to estimate the OFRs water volume. However, for the modeling scenarios, the OFRs water volume changes according to their surface area, and therefore, it is not kept constant, and modeled according to Equation 3.

**Reference**

Yaeger et al: Trends in the construction of on-farm irrigation reservoirs in response to aquifer decline in eastern is not well referenced

**Correct Reference:**

Yaeger, M. A., Massey, J. H., Reba, M. L., and Adviento-Borbe, M. A. A.: Trends in the construction of on-farm irrigation reservoirs in response to aquifer decline in eastern, Agric. Water Manag., 208, 373–383, https://doi.org/10.1016/j.agwat.2018.06.040, 2018.

**#Reviewer2**

Dear Authors,

The study innovatively assessed the spatial and temporal variability of the cumulative impact of OFRs at the watershed and subwatersheds levels, quantifying the annual impacts of the OFRs on flow and peak flow at the channel scale. Although the manuscript presents relevant novelty, there are some issues, which must be addressed or clarified by the Authors, prior to the study's publication, in my point of view. Please, see below my comments and suggestions.

**Major comments:**

- The influence of OFRs on surface hydrology and their mathematical representation could not be adequately evaluated, because there is no streamflow monitoring in the areas with the presence of the OFRs (see Fig. 1).
  - Hopefully, this issue is clarified with the information added in "Section 2.4 OFRs water balance (see lines 281-324)".
- 2. Very simple considerations were assumed for the water volume and water releases of OFRs (see Lines 273-278). This is a critical issue of the study, which should be addressed by fieldwork or at least by a sensitivity analysis.

True, and an important limitation. We added a few more lines describing this limitation.

See added lines 291:294:

"This is a significant limitation of our study, as the assumption was necessary due to the absence of available bathymetry data."

This was already discussed in lines 708:718:

"Future improvements should focus on how to better represent the OFRs water management (i.e., OFRs inflows and outflows) in SWAT+. Given that each OFR has an independent water balance, accounting for the OFR water volume change would be a more realistic representation of the OFRs when compared to the three surface area scenarios tested in this study. Estimating the OFRs volume change can be done by combining the OFRs surface area time series with area-elevation equations—these

equations describe the bathymetry of the OFRs, and allow volume estimation by inputting the OFRs' surface area (Liebe et al., 2005; Meigh, 1995; Sawunyama et al., 2006). After carefully assessing different methods to derive these equations (Arvor et al., 2018; Avisse et al., 2017; Li et al., 2021; Meigh, 1995; Sawunyama et al., 2006; Vanthof & Kelly, 2019; Yao et al., 2018; Zhang et al., 2016), we decided that measured ground-data of the OFRs' depth—which is not available—is required to estimate the equations with an acceptable uncertainty"

3. Please, explain in detail how hydrologically the OFRs could positively impact streamflow, as you found.

This is related to the analysis being carried out at the channel level, and it was observed for a few months of the year, based on how the water balance is calculated.

Clarified this issue by adding information to lines 603:613:

"When evaluating flow changes at the channel scale, it's important to note that flow at this level is several orders of magnitude smaller than at the main basin outlet. Consequently, this scale often exhibits more significant percentage changes, both increases and decreases. This likely explains how OFRs can enhance channel flow, primarily due to the additional water contributed by OFRs, influenced by periods of increased precipitation in certain channels during specific months and years. While we calculated the monthly impact on flow at the channel scale by aggregating the OFRs to the closest channel, previous studies have mostly reported the annual impact on flows (Habets et al., 2018), and they performed their analysis at the subwatershed scale by aggregating the OFRs to a single point at the outlet of each subwatershed in SWAT (Evenson et al., 2018; Kim & Parajuli, 2014; Perrin, 2012; Zhang et al., 2012), or they used different modeling approaches (see Habet et al., (2018))"

4. I did not fully understand what kind of results you would like to explore in section 3.5 Overall impact of OFRs. Please, improve this section.

The spatial variability of the impact of the OFRs. We changed the section name.

"Section 3.5 Spatial variability of the OFRs impact on annual flow".

Added extra information to lines 677:680 to clarify this statement:

"Our results indicate that OFRs do not have an equally distributed impact on mean and peak flow across the watershed. This variability is primarily influenced by differences in their size, water storage capacity, and their spatial distribution (i.e., their presence)".

5. In the beginning of the manuscript, you mentioned that one of your focuses is the analysis of the interannual flow variability. However, I could not find such an analysis. An example of simulated time series (section 3.4) is not enough. For example, what are the impacts of OFRs on low-flow and high-low years?

Figure 5 highlights monthly changes and during all simulated years. For example, for the month of January, we analyzed how the OFRs contributed to changes in channel flow, using data from January between 1990 and 2020 for all OFRs. This variability, which also includes years with low and high flows, is shown by the size of the bars for each month.

To clarify this issue, we added an extra sentence to the Figure 5 caption:

"This analysis included data from all simulated years (1990-2020)."

**comments:**

1. Lines 385-386: "In general, the OFRs contributed to decreased monthly flow. However, the OFRs' impact on flow had a significant intra- and inter-annual variability..." Figure 5 is only on intra-annual variability or, also called, seasonal variability.

A close examination of Figure 5 reveals its various components. The channels are categorized into four classes, and for each class, we assess how the flow fluctuated over the course of the year, represented along the x-axis by each month. The three bar colors indicate three distinct scenarios, while the bar heights reflect differences among channels as well as variations across years. For example, when examining the bars for January, they encompass all January data from 1990 to 2020.

Added extra information to lines 425:429:

"Figure 5 breaks down the channels into four distinct categories, with each category showing variations in flow throughout the year, displayed along the x-axis by month. The three bar colors represent different scenarios, while bar heights illustrate variations across channels and years. For example, the bars for January include all January data spanning from 1990 to 2020, enabling a thorough comparison of seasonal and year-to-year flow changes."

---

## Author Response (AR2)

We thank the additional reviewer for their constructive comments, which we addressed in blue below and incorporated into our manuscript. We would also like to thank the editorial team for their thoroughness. However, our paper has been under review with HESS for over a year, during which we have addressed two rounds of revisions. It would be appreciated if a resolution could be reached soon.

I have read this paper with some interest. However, I don't think it is as novel as the authors state. While the authors state that they are not aware of any other work that combines the spatial and temporal ability of OFRs with multi-satellite imagery, I would argue that the paper by Robertson and the associated work on assessing the volume of OFRs is probably more advanced than this study. There is similar work by Xie Yan et al. 2023. (See references below)

We removed the word "novel" in most instances and fully acknowledge the limitations of our work: "It is important to keep in mind that the impact of the OFRs on this study is solely based on modeling scenarios and does not account for OFR management practices, which represents a key limitation of this simulation study."

Thank you for pointing out the studies by Robertson et al. (2023) and by Xie Yan et al. (2023). While we acknowledge the relevance of these studies, they try to understand the problem through a different lens without incorporating on-farm reservoir surface water extent derived based on remotely sensed data in their hydrological model, which we believe is the new perspective we are providing here.

Robertson et al. (2023) quantified how the interaction between climate change and farm dams affects streamflow characteristics. They relied on the farm dam dataset produced in Malerba et al. (2021), which includes point data for farm dams, rather than their varying surface water extent (like in our work presented here). Their use of remote sensing was a straightforward method, using annual time-series of water index thresholds, on three Landsat sensors to determine the first year where water is observed in each farm dam from Malerba et al. (2021), which was then used as a proxy for the year of dam construction in their modeling.

Yan et al. (2023) employed a hydrological model to distinguish the impacts of climate variability, land-use change, and small- to medium-sized reservoirs on streamflow in two river basins in Southeast China.

Their use of remote sensing was ONLY in visually deriving the land use data based on Landsat 5 data. The obtained the reservoir data from one of the local provinces where their study catchments are located (quote taken directly from their paper: "In addition, there were also reservoir data during the study period (1970–2015), including the location, scale (catchment area, storage capacity), type, and construction time, which were obtained from Water Resources Management Department of Fujian Province").

We reviewed and cited the two studies in our manuscript. The new lines in the manuscript are copied below: "For example, other studies have found that OFRs reduce annual and monthly runoffs in southeastern China (Yan et al. 2023) and Australia's Murray-Darling Basin (Robertson et al. 2023)."

The large body of work on this topic generally highlights the major difficulties and uncertainties in assessing the impact of OFRs.

This paper identifies those too, but ultimately they only address one of the major sources of uncertainty i.e the model structural uncertainty in representing spatial and temporal variability of OFRs.

The other major sources of uncertainty identified in the literature i.e surface area volume relationships of the systems and the management thereof for irrigation or other abstractions.

We fully agree with the reviewer and provide the reasons why we were ultimately able to address only one of the sources of uncertainty.

We have not done so for a few reasons:

- Irrigation data is rarely available
- For this type of irrigation (from OFRs), farmers do not have meters on their reservoirs—even if they have meters, the data would be sparse. Hence, calculating water abstract was unfortunately not feasible.
- Surface area-volume relationships are helpful, but also limited, given that the OFRs do not follow a consistent construction pattern and vary from farm to farm. A rough estimate using a trapezoidal equation would probably be as accurate as possible if you do not measure the bathymetry in situ.

This is expressed in the manuscript in the following paragraphs:

"We did not have access to water abstraction data from the OFRs, so all abstractions were modeled using Equation 3, which accounts for water flowing out of the OFR, as well as losses from evaporation and seepage. The total volume of water in the OFR fluctuates in response to changes in surface area and is also influenced by evaporation losses and the operation of the spillway. A reduction in surface area (Equation 4) typically leads to a corresponding decrease in water volume. If inflows are insufficient to fill the OFR, water will not be routed to the downstream channel."

"For each of the aggregated OFR, the initial water volume (Vstored, see Equation 3) was calculated using the SWAT+ default rule, which is a simple multiplication of the OFR surface area by a factor of 10, similar to other studies based on SWAT+ (Ni and Parajuli, 2018; Zhang et al., 2012). For a scenario where the OFR has a surface area of 1 hectare (10,000 m²), the corresponding volume would be 100,000 m³—this is a limitation of our study, as the assumption was necessary due to the absence of available bathymetry data. In addition, since we did not have access to the OFRs' release rates, we used the model's default release rule, which sets the OFRs to release water when the spillway volume is reached—80% of the OFRs' capacity (Bieger et al., 2017)."

• The uncertainties related to the spatial distribution of the reservoirs and how to capture those in hydrological modeling.

o Although I'm not familiar with the latest version of SWAT/SWAT+, based on the paper, it seems that recent advances do provide a useful way of capturing the spatial distribution of reservoirs and simulating their impact.

We have conducted our modeling at the channel level, which incorporates more of the spatial variability in reservoir distribution, illustrated in the following paragraph: "In addition, the latest versions allow for adding more than one OFR per subwatershed by associating the OFR with channels—components of the watersheds, as well as finer divisions and extensions of water stream reaches, enabling modeling analyses at the channel scale. When simulating the impact of the OFRs at the channel scale, there is a higher level of detail of where and when the OFRs are contributing to changes in surface hydrology, unlike the previous versions of the model, which allowed adding only a single OFR per subwatershed placed at the subwatershed outlet as a point (Arnold et al., 2012), and therefore, the analyses were conducted at the subwatershed scale."

• The difficulty in obtaining and estimating the volume associated with small farm dams and their identification.

o To some extent, the authors address this issue in the paper. However, I do not think that they do so adequately. In effect, they conclude that measured ground data of OFR depth is needed . However, various tools that have been developed globally to address this problem.

For OFRs of the small size we are working with, to the best of our knowledge, there are no accurate ways of doing this, without ground measurements, which we did not have access to. Given that these are small water bodies, a rough trapezoidal equation would likely provide a very *rough* estimation, using a local DEM. But given the small size of OFRs, you can miss the total water volume by a lot, even when using local DEMs (e.g., by 100%). We addressed this in the paper in the following paragraphs:

"Estimating the OFR's volume change can be done by combining the OFR surface area time series with area-elevation equations—these equations describe the OFR's bathymetry, and allow volume estimation by inputting the OFR's surface area (Liebe et al., 2005; Meigh, 1995; Sawunyama et al., 2006). After carefully assessing different methods to derive these equations (Arvor et al., 2018; Avisse et al., 2017; Li et al., 2021; Meigh, 1995; Sawunyama et al., 2006; Vanthof & Kelly, 2019; Yao et al., 2018; Zhang et al., 2016), we concluded that measured ground data of the OFRs' depth—which is not available—is required to estimate the equations with an acceptable uncertainty. Estimating the area-elevation equations entails several challenges, including: 1) even though there are several DEMs available for the study region (Arkansas GIS Office, 2022)—DEMs can be used to estimate the OFRs bottom elevation—the DEMs were collected when most of the OFRs were full (i.e., bathymetry was not exposed), which limits their use in this case; and 2) although the OFRs are located within the same geomorphological region, they have different depth, shape and physical characteristics (Perin et al., 2022; Yaeger et al., 2017). Therefore, even if a generalized area-elevation equation were calculated for our study region—this is a common approach employed by other studies (Mady et al., 2020; Vanthof and Kelly, 2019)—that would still lead to high uncertainties in water volume changes."

"Future improvements should focus on how to better represent OFR's water management (i.e., OFR's inflows and outflows) in SWAT+. Given that each OFR has an independent water balance, accounting for the OFRs water volume change would be a more realistic representation of the OFRs when compared to the three surface area scenarios tested in this study. Estimating the OFRs volume change can be done by combining the OFR surface area time series with area-elevation equations—these equations describe the OFRs' bathymetry, and allow volume estimation by inputting the OFRs' surface area (Liebe et al., 2005; Meigh, 1995; Sawunyama et al., 2006). After carefully assessing different methods to derive these equations (Arvor et al., 2018; Avisse et al., 2017; Li et al., 2021; Meigh, 1995; Sawunyama et al., 2006; Vanthof & Kelly, 2019; Yao et al., 2018; Zhang et al., 2016), we decided that measured ground-data of the OFRs' depth-which is not available-is required to estimate the equations with an acceptable uncertainty. Estimating the area-elevation equations entails several challenges, including: 1) despite the fact that there are several DEMs available for the study region (Arkansas GIS Office, 2022)—DEMs can be used to estimate the OFRs bottom elevation—the DEMs were collected when most of the OFRs were full (i.e., bathymetry was not exposed), which limits their use in this case; and 2) although the OFRs are located within the same geomorphological region, they have different depth, shape and physical characteristics (Perin et al., 2022; Yaeger et al., 2017). Therefore, even if a generalized area-elevation equation were calculated for our study region—this is a common approach employed by other studies (Mady et al., 2020; Vanthof and Kelly, 2019)—that would still lead to high uncertainties in water volume changes. Ideally, each OFR would have its own equation, which was not possible when this study was done. Future work should integrate data on

actual evapotranspiration, ET (Kiptala et al., 2014), to quantify the balance between water availability and ET, which determines, in large part, the irrigation system efficiency and crop productivity in the watersheds where OFRs occur."

In particular, the work in Australia, which was based on a large sample of measured reservoirs, is relevant (Malerba et al., 2021). In essence, the authors provide a rebuttal of why they have not addressed this aspect - but in essence, it does not advance the understanding in how to address a major area of uncertainty in the study.

We believe the main contribution of our work is the integration of remotely sensed surface water extent of OFRs in SWAT+. We agree that Malerba's work is great, but they do have access to ground measurements, which unfortunately we did not have access to.

• The major uncertainty around OFRs is how they are managed and how much water for irrigation is actually abstracted from them. Other studies (e.g., Hughes and Mantel) identified this as the major source of uncertainty in such studies - greater than model structure and spatial representation type uncertainties.

o I find this aspect a little bit odd in the paper. The authors identify the importance of OFRs for irrigation in the justification for the paper (Lines 18, 20, 42-43) and recognize that how they are managed has a particular impact on surface water hydrology. However, they never consider irrigation in the model configuration. Section 2.2 deals only with the SWAT+ setup to simulate OFRs but completely neglects any management of the OFRs, such as irrigation abstractions.

We fully agree with the reviewer, however, as mentioned before, we have not done so because this data was not available.

Even if we had access to this data, translating it so that SWAT+ would take those measurements would involve model modifications, which is outside the scope of this work.

Our intention was to model the water bodies as part of the hydrological systems. I.e., what is the impact of their presence?

Having irrigation data would be a plus, but we unfortunately did not have access to such data.

o Ultimately, the authors conclude that this is still a major area of uncertainty and make some recommendations for the enhancement of the SWAT model to deal with this. In essence, the known or estimated inflows and outflows cannot be accommodated in the current form and that the "decision tables" approach of SWAT is too difficult to set up for multiple reservoirs. They do not make new recommendations about how this could be improved. Line 166 highlights the new flexibility in defining management schedules - but the paper does not address this. Had they really got to grips with this aspect, this would provide a strong innovative aspect of the paper.

There is a clear gap that needs to be addressed in SWAT, but we are not SWAT+ modelers per-se and thus our suggestions are primarily centered around the data needs and availability rather than the details in the model. However, we have made some modeling suggestions in the manuscript:

"Efforts should also be made to improve SWAT+ capabilities to receive measured OFRs' inflows and outflows. The latest version of the model has improved the hydrological representation of small water impoundments in SWAT+ (Mollina-Navarro et al., 2018). Nonetheless, at the time of our study, the newest version of the model does not allow users to input measured or calculated OFRs' inflows and outflows. Instead, the model developers recommend simulating the OFRs water balance using decision tables (Arnold et al., 2018; Dile et al., 2022). However, there are very limited guidelines on how to

create these decision tables. In addition, the tables would simulate the OFRs water balance instead of using the measured or calculated volume change, which could introduce more uncertainties to the modeling scenarios."

Ultimately, this is a useful technical study combining a number of different techniques, i.e., remote sensing and application of SWAT+ to provide an effective spatial representation of OFRs and a useful framework in which this is done. However, the paper does not really advance this field significantly because the major source of uncertainty, i.e., irrigation abstraction and other management-related aspects of the reservoirs, are completely neglected,

It is not clear whether this is because the model cannot represent them, a lack of data, or other reasons. It is still a major shortcoming in the paper and needs to be better acknowledged as such. Whether the paper is then publishable or not falls back to how unique the combination of the satellite remote sensing coupled to the process-based hydrological model really is.

We thank the reviewer for this comment. We are aware of this shortcoming, which is due to the lack of data availability in our case. We expressed this in the paper in the future improvements section.

Our main improvements are summarized in 4.3 Research implications and applications to other study regions and several other paragraphs: "Studies have employed either data-driven or physically based hydrological model approaches to estimate the effects of OFRs on watersheds. However, combining these approaches provides a better understanding of the spatial and temporal variability of OFR impacts, as it incorporates the dynamic changes of OFRs into the hydrological model. Studies have either used data driven or physically based hydrological model approaches to estimate OFR impacts on the watersheds, despite the fact that combining the two approaches leads to better understanding on what is the spatial and temporal variability of the OFR impacts, given that the dynamic changes of the OFRs are incorporated into the hydrological model. To quantify whether the impact of the OFRS on mean and peak flow varies intra- and inter-annually, and which subwatersheds are more affected, we combined a data-driven remote sensing-based model with the latest improvements in SWAT+ to assess the OFR impacts."

And in this paragraph: "By leveraging the latest improvements in SWAT+ to simulate water impoundments (Molina-Navarro et al., 2018) and combining them with a novel algorithm based on time series of satellite data to monitor OFRs (Perin et al., 2022), we modeled the impact of OFRs on flow at the channel scale. In addition, the surface area scenarios enabled us to account for events when the OFRs were at the lowest, regular, and fullest capacities according to their surface area (see Fig. 2). This is an improvement over previous studies (e.g., Ni et al., 2020; Ni and Parajuli, 2018; Perrin, 2012) that used a single surface area (i.e., one snapshot in time) to represent the OFRs in SWAT."

M.E. Malerba, N. Wright, P.I. Macreadie A Continental-Scale Assessment of Density, Size, Distribution and Historical Trends of Farm Dams Using Deep Learning Convolutional Neural Networks Remote Sens. (Basel), 13 (2) (2021), 10.3390/rs13020319

Hughes, D. A. & Mantel, S. K. (2010) Estimating the uncertainty in simulating the impacts of small farm dams on streamflowregimes in South Africa. Hydrol. Sci. J. 55(4), 578–592.

David E. Robertson, Hongxing Zheng, Jorge L. Peña-Arancibia, Francis H.S. Chiew, Santosh Aryal, Martino Malerba, Nicholas Wright, How sensitive are catchment runoff estimates to on-farm storages under current and future climates?, Journal of Hydrology, Volume 626, Part A, 2023

Xie Yan, Bingqing Lin, Xingwei Chen, Huaxia Yao, Weifang Ruan, Xiaocheng Li, Impacts of small and medium-sized reservoirs on streamflow in two basins of Southeast China, using a hydrological model to separate influences of multiple drivers, Journal of Hydrology: Regional Studies, Volume 50, 2023,

Ignacio Fuentes, Richard Scalzo, R. Willem Vervoort, Volume and uncertainty estimates of on-farm reservoirs using surface reflectance and LiDAR data, Environmental Modelling & Software, Volume 143, 2021,"

---

## Author Response (AR3)

**Deadline 31 Aug**

Public justification (visible to the public if the article is accepted and published): (The line numbers quoted below refer to the track changes version of the revised manuscript.)

We thank the editor for their constructive comments, which we addressed in blue below and incorporated into our manuscript. We would also like to thank the editorial team for their thoroughness. However, our paper has been under review with HESS for over a year (since May 2024), during which we have addressed three rounds of revisions. We would appreciate a resolution being reached soon.

I find that the authors did not adequately respond to a major concern of the 3rd reviewer, namely

"The major uncertainty around OFRs is how they are managed and how much water for irrigation is actually abstracted from them. [...] I find this aspect a little bit odd in the paper. The authors [...] never consider irrigation in the model configuration." And again: "... the paper does not really advance this field significantly because the major source of uncertainty, i.e., irrigation abstraction and other management-related aspects of the reservoirs, are completely neglected"

The premise of the paper is that OFRs impact the hydrology (e.g. lines 51-52). But the main impact of OFRs on the hydrology is not the OFR itself but rather how the OFR is being managed, namely for using the stored water for irrigation. Without the use of the stored water in OFRs for irrigation, OFRs will have hardly a noticeable impact on the hydrology. But as this irrigation use of the stored water is ignored in the model used, the innovation of this paper is limited.

For example, there might be a correlation between the change in the number or size of OFRs in a certain area and a change in the irrigated area in that same area, and with it the change in water consumption by that change in irrigated area. There are currently many RS products that can reliably estimate such changes in water consumption patterns. Including this could possibly significantly improve our understanding of the impact of OFRs on the hydrology. I write this just to invite the authors to more adequately reflect on the 3rd reviewer's concern.

Thank you for pointing this out. The reasons why we did not include irrigation from OFRs are the following: (1) Most irrigation withdrawals occur directly from the channel (e.g., USDA 2023), so the withdrawal term is excluded in Equation 3 (Please also see the answer to #3 below). (2) Given that OFRs are on-farm small water bodies, very few farmers keep track of the irrigation data at the individual OFR level, and water abstraction is not resolved by typical RS data products, and (3) Lastly, our method of comparing naturalized flows (i.e., baseline flows without

any OFRs or withdrawals) with observed inflows into OFRs, allows us to quantify how OFRs modify the natural flow regime, similar to other studies (Döll et al. 2009).

Döll, P., Fiedler, K., and Zhang, J.: Global-scale analysis of river flow alterations due to water withdrawals and reservoirs, *Hydrol. Earth Syst. Sci.*, 13, 2413–2432, <a href="https://doi.org/10.5194/hess-13-2413-2009">https://doi.org/10.5194/hess-13-2413-2009</a>, 2009.

USDA, 2023. Irrigation Organizations: Water Inflows and Outflows: https://ers.usda.gov/sites/default/files/ laserfiche/publications/107067/EB-36.pdf?v=48646

I have a few more detailed comments on the revised manuscript, as follows, which I would like the authors also to address:

1. In your rebuttal you promised the following, but I could not find this in the revised manuscript:

"We reviewed and cited the two studies in our manuscript. The new lines in the manuscript are copied below: "For example, previous studies have found that OFRs reduce annual and monthly runoffs in southeastern China (Yan et al. 2023) and Australia's Murray-Darling Basin (Robertson et al. 2023)."

Added the text above and citations.

2. In lines 58-59 you write "... the spatial and temporal variability of OFRs ..." What do you mean by this? Kindly clarify.

We clarified by replacing it with "the spatial and temporal variability of surface water extent in OFRs."

3. The paper frequently refers to inflows and outflows of OFRs (e.g. line 74, 568-569, 639, 692, 716, 719). But farmers also abstract water from OFRs through e.g. pumps and pipes, in order to irrigate their crops. This abstraction is also not covered in the OFR water balance given in the new equation 3 (line 294). Kindly explain why you omit this, in view important, feature of OFRs.

We thank the editor for this point. While farmers may abstract water from OFRs for irrigation through pumps or pipes, in our modeling framework, we assume that the main irrigation withdrawals in the study area are taken directly along the channels and streams, not from the small OFRs themselves. This assumption is supported by hydrological evidence that irrigation withdrawals predominantly occur at the channel level in many agricultural systems (e.g., Brochet et al., 2024). Hence, we did not include an explicit irrigation abstraction term in the OFR water balance (Equation 3). Inflows into OFRs already reflect upstream abstractions, and therefore, the observed inflow signal implicitly accounts for this impact.

We acknowledge this as a limitation of our approach: direct withdrawals from OFRs by individual farmers are not explicitly modeled, both due to the absence of reservoir-level

irrigation data and because OFRs are typically small water bodies, making it infeasible to quantify such abstractions reliably with existing datasets or remote sensing products. We have clarified this assumption and its implications in the revised text, by including the following:

"While farmers may occasionally withdraw water directly from OFRs, in our study region, most irrigation appropriations are taken from channels and streams. This is consistent with irrigation practices in Arkansas, where large-scale surface water projects withdraw directly from rivers and distribute water via canals and pipelines. Similarly, watershed-scale modeling that incorporates irrigation withdrawals into the river system yields better flow simulations, especially during low-flow periods (Brochet et al., 2024). Given this, and in the absence of high-resolution data on reservoir-specific withdrawals, our framework assumes that inflow to OFRs already reflects upstream irrigation abstractions. Thus, Equation 3 omits an explicit irrigation withdrawal term for OFRs, and our approach focuses on quantifying the hydrological signal alterations through natural versus reservoir-influenced flow comparisons".

Brochet, M., Raimonet, M., Le Moine, N., Ducharne, A., Cheruy, F., Ottlé, C., Ducharne, F., and Gascoin, S.: Accounting for irrigation water withdrawals improves low-flow simulations of a regional watershed model, Hydrol. Earth Syst. Sci., 28, 49–69, https://doi.org/10.5194/hess-28-49-2024, 2024.

4. In line 280-281 the paper now reads: "The total volume of water in the OFR fluctuates in response to changes in surface area ..." I think this is an incorrect statement. The volume of water in an OFR changes due to changes in inflows, outflows, or abstractions. Such changes of water volumes are associated with changes in the surface area of the OFR, but are not caused by it, as is suggested in this text.

We rephrased it to the following, as suggested: "The volume of water in an OFR, which changes due to changes in inflows, outflows, or abstractions, is associated with changes in the surface area of the OFR".

5. Lines 311-314: I do not understand what the volume of water and the surface area at the emergency spillway or at the principal spillway mean. For readers that are unfamiliar with the specific design of an OFR, like myself, this needs an explanation.

We added the following explanation to the text: "Spillways release the water once it reaches a specific level. Most OFRs have uncontrolled spillways, implying that there are no gates to control the outflow. The outflow through the spillway depends on the level above the spillway crest. An emergency spillway, whose crest is typically at a higher elevation than the principal spillway, is an additional spillway designed to release excess water during heavy flooding. The surface area of the OFR represents the water spread area corresponding to a given level in the reservoir, which typically increases as the reservoir level rises."

6. Line 352: "SWOT+" Is this correct? Changed to SWAT+

- 7. Lines 557-560: Why does this sentence start with "Although"? In my understanding the sentence could more appropriately start with the word "As".

  Changed to start with "As" as suggested.
- 8. Line 641: "... may not lead to differences (e.g., > 10%) ..." I would add: "significant", as follows: "... may not lead to significant differences (e.g., > 10%) ..."

  Added as suggested.
- 9. At lines 184 and 735 you refer to Mollina-Navarro et al., but that should read Molina-Navarro et al.

Changed as suggested.